# Back to Square Roots: An Optimal Bound on the Matrix Factorization Error for Multi-Epoch Differentially Private SGD

**Nikita P. Kalinin**
Institute of Science and Technology (ISTA)
Klosterneuburg, Austria
`nikita.kalinin@ist.ac.at`

**Ryan McKenna**
Google Research
`mckennar@google.com`

**Jalaj Upadhyay**
Rutgers University
New Jersey, USA
`jalaj.upadhyay@rutgers.edu`

**Christoph H. Lampert**
Institute of Science and Technology (ISTA)
Klosterneuburg, Austria
`chl@ist.ac.at`

## Abstract

Matrix factorization mechanisms for differentially private training have emerged as a promising approach to improve model utility under privacy constraints. In practical settings, models are typically trained over multiple epochs, requiring matrix factorizations that account for repeated participation. Existing theoretical upper and lower bounds on multi-epoch factorization error leave a significant gap. In this work, we introduce a new explicit factorization method, Banded Inverse Square Root (BISR), which imposes a banded structure on the inverse correlation matrix. This factorization enables us to derive an explicit and tight characterization of the multi-epoch error. We further prove that BISR achieves asymptotically optimal error by matching the upper and lower bounds. Empirically, BISR performs on par with state-of-the-art factorization methods, while being simpler to implement, computationally efficient, and easier to analyze.

## 1 Introduction

Private machine learning has become increasingly important as the use of sensitive data in model training continues to grow. Ensuring privacy while maintaining model accuracy presents a critical challenge, particularly in fields like healthcare, finance, and personal data analysis. *Differential Privacy* (DP) has emerged as a fundamental framework for formalizing privacy guarantees in machine learning. It provides a mathematically rigorous way to limit the influence of any individual data point on the model's output, thereby preserving privacy. One effective approach to achieving DP in iterative training is through the use of structured noise mechanisms that balance privacy guarantees with model utility.

In this work, we focus on the *Matrix Factorization Mechanism* (see Section 2.1 for a formal description) for ensuring DP. Matrix factorization has been extensively studied in recent years in the broader machine learning application (Kairouz et al., 2021; Denisov et al., 2022; Fichtenberger et al., 2023; Henzinger et al., 2023; Choquette-Choo et al., 2023b; Andersson & Pagh, 2023; Henzinger et al., 2024; Kalinin & Lampert, 2024; Henzinger & Upadhyay, 2025; Pillutla et al., 2025; Henzinger et al., 2025b). It has also been investigated in private optimization (Koloskova et al., 2023; Choquette-Choo et al., 2024a), federated learning (Zhang et al., 2025; Bienstock et al., 2025), and in the context of adaptive optimizers (Kalinin et al., 2025; Ganesh et al., 2025). In particular, memory-efficient matrix factorization has attracted significant attention in recent years (McMahan et al., 2024; Dvijotham et al., 2024; Andersson & Pagh, 2025; Henzinger et al., 2025a).

The approach has also been adopted in practice; for example, Google has reported its use for training production on-device language models in their 2024 blog post "Advances in private training for production on-device language models" (Xu & Zhang, 2024). The core idea of the MF mechanism

is to inject correlated noise into the gradients during training. The correlations are determined by the inverse of a matrix $C \in \mathbb{R}^{n \times n}$, referred to as the *strategy matrix*. While $C$ serves as a factor in the matrix factorization that defines the mechanism and used in calculation of the privacy level, its inverse $C^{-1}$ functions as the *correlation matrix*, specifying how the injected noise is correlated across training steps. This structure allows the mechanism to preserve model accuracy while still guaranteeing privacy.

Intuitively, the mechanism can be seen as follows: at each training step, fresh noise is generated and added, but part of the previous noise is stored in a buffer. In subsequent steps, portions of the stored noise are subtracted in a controlled manner. This cancellation effect reduces the total amount of noise that accumulates in the model, thereby improving utility without weakening privacy guarantees.

When computing privacy, we must account for *multi-epoch participation*, since in multi-epoch training the same datapoints are used multiple times. The notion of multi-epoch participation in the context of matrix factorization was first introduced by Choquette-Choo et al. (2023a), where it was formulated as an optimization problem over banded matrices. However, a key limitation of existing methods is the lack of precise theoretical guarantees on the *factorization error* in multi-epoch participation. While Kalinin & Lampert (2024) established a general lower bound and provided an upper bound for *Square Root Factorization*, the error bounds for *Banded Square Root Factorization*, where the correlation matrix $C$ is $p$-banded, remained imprecise with respect to the bandwidth $p$.

In this work, we propose a novel approach to matrix factorization: rather than imposing a banded structure on the correlation matrix $C$, we introduce a *banded inverse square root*, enforcing the banded structure on $C^{-1}$. This shift[1] offers several key advantages. First, it allows for precise control over the resulting factorized matrices, enabling us to derive **explicit upper bounds** on the factorization error with clear dependence on the bandwidth. Second, the method is **computationally efficient**, as it requires one just to convolve the previous noise with a quickly computable fixed sequence of coefficients, which can be done for instance via Fast Fourier Transform (FFT), making it suitable for large-scale machine learning tasks. Most importantly, we prove that our method achieves **asymptotically optimal factorization error**: we establish a **new lower bound** that matches our upper bound, closing a significant theoretical gap in the literature.

By refining the theoretical understanding of banded factorization in multi-epoch settings, our work provides both theoretical insights and practical benefits for privacy-preserving ML training. Our main contributions are:

1. We introduce a new factorization method, the ***Banded Inverse Square Root (BISR)***, which is scalable, efficient, and agnostic to the underlying training objective.
2. We prove that BISR is **asymptotically optimal**, by deriving tight upper and lower bounds on the multi-epoch factorization error, with explicit dependence on bandwidth and workload properties.
3. We conduct a thorough empirical evaluation, comparing BISR to existing techniques in multi participation trainingincluding Banded Square Root (BSR) (Kalinin & Lampert, 2024), Buffered Linear Toeplitz (BLT) (Dvijotham et al., 2024), and Banded Matrix Factorization (Band-MF) (McKenna, 2025), showing that BISR achieves a higher or comparable accuracy for the large matrix sizes.
4. In the low-memory regime, we propose an optimization method, **BandInvMF**, which directly optimizes the coefficients of the matrix $C^{-1}$. This approach achieves error rates comparable to state-of-the-art factorization methods, while being easy and efficient to implement.

## 2 BACKGROUND

### 2.1 MATRIX FACTORIZATION (MF)

MF mechanisms provide a promising approach to the private matrix multiplication problem, which has applications in continual counting and Stochastic Gradient Descent for machine learning. Specifically, we aim to estimate the product of a public matrix of coefficients $A \in \mathbb{R}^{n \times n}$ and a private matrix $X \in \mathbb{R}^{n \times d}$. Instead of doing so directly, we adopt a factorization $A = BC$, allowing us to estimate

---

[1]The inverse correlation matrix has been receiving more attention recently. In the concurrent work McMahan & Pillutla (2025), the authors consider the inverse correlation matrix of BLT.

$AX$ privately as $\widehat{AX} = B(CX + Z) = A(X + C^{-1}Z)$. Here, $Z \sim \mathcal{N}(0, s^2)^{n \times d}$ is appropriately scaled Gaussian noise, which ensures that $CX + Z \in \mathbb{R}^{n \times d}$ is private; the multiplication by $B$ preserves the privacy guarantees due to DP's post-processing property.

The choice of factorization $A = BC$ can significantly impact the quality of the private estimation. We quantify the *approximation quality* by the expected Frobenius error of the estimated product,

$$\mathcal{E}(B, C)^2 = \frac{1}{n}\mathbb{E}_Z \|AX - \widehat{AX}\|_{\mathrm{F}}^2, \tag{1}$$

where $\| \cdot \|_{\mathrm{F}}$ is the Frobenius norm. An elementary analysis (Li et al., 2015) shows that

$$\mathcal{E}(B, C)^2 = \frac{s^2}{n}\|B\|_{\mathrm{F}}^2, \tag{2}$$

and that the required noise strength, $s$, scales proportionally to the *sensitivity* of the matrix $C$. Let $X \sim X'$ indicate that the update vector sequences differ only in the entries corresponding to a single data item. Then the sensitivity of the matrix $C$ is defined as

$$\mathrm{sens}(C) := \sup_{X \sim X'} \|CX - CX'\|_{\mathrm{F}} \tag{3}$$

**Private SGD.** In this work, we consider the task of model training with SGD with (optional) weight decay and momentum. The corresponding update equations are $\theta_{i+1} = \alpha\theta_i - m_{i+1}$ and $m_{i+1} = \beta m_i + x_i$, where $\theta_1, \ldots, \theta_n \in \mathbb{R}^D$ are the model parameters after each update step, $x_1, \ldots, x_n$ are the gradient vectors computed in each update step, $0 < \alpha \le 1$ is the weight decay factor, and $0 \le \beta < 1$ is the momentum strength [2].

Following Kalinin & Lampert (2024), we rewrite the dynamics in the matrix form as $\Theta = A_{\alpha,\beta}X$, with $\Theta = (\theta_1, \ldots, \theta_n)^\top \in \mathbb{R}^{n \times D}$, $X = (x_1, \ldots, x_n)^\top \in \mathbb{R}^{n \times D}$, and $A_{\alpha,\beta}$ is the *SGD workload matrix* defined as follows:

$$A_{\alpha,\beta} = \begin{pmatrix} 1 & 0 & \cdots & 0 \\ \alpha + \beta & 1 & \cdots & 0 \\ \vdots & \vdots & \ddots & \vdots \\ \sum_{j=0}^{n-1} \alpha^j \beta^{n-1-j} & \sum_{j=0}^{n-2} \alpha^j \beta^{n-2-j} & \cdots & 1 \end{pmatrix} \in \mathbb{R}^{n \times n}. \tag{4}$$

Note that, in contrast to the naive MF setting, in the SGD case any input data (gradient) $x_i$ depends on the previously computed model parameters, $\theta_{i-1}$, that is, we aim for *adaptive privacy*. However, Denisov et al. (2022) shows that for Gaussian noise, adaptive privacy follows from the non-adaptive one, i.e., it suffices for us to solve the case in which the $X$ matrix is an arbitrary fixed data matrix. Consequently, we estimate $A_{\alpha,\beta}X$ privately using the form $\widehat{A_{\alpha,\beta}X} = A_{\alpha,\beta}(X + C^{-1}Z)$. This form corresponds to running SGD, but each individual gradient update is perturbed by a correlated noise vector. That has the advantage that we do not need to store any previous gradients, and we can rely on any existing implementation of the SGD procedure.

In multi-epoch SGD, each data sample might contribute to more than one gradient update vector. As a suitable notion of sensitivity, we adopt the setting of $b$-min-separated repeated participation (two participations of any data point occur at least $b$ update steps apart). The resulting sensitivity can be bounded as Choquette-Choo et al. (2023a):

$$\mathrm{sens}_{k,b}(C) \le \max_{\pi \in \Pi_{k,b}} \sqrt{\sum_{i,j \in \pi} |(C^\top C)_{[i,j]}|} \tag{5}$$

where $\Pi_{k,b}$ denotes the collection of index sets drawn from $\{1, \ldots, n\}$ that contains at most $k$ elements and satisfy the condition that any two distinct indices are separated by at least $b$ positions, so that no two indices in the set lie closer than $b$ apart.

This bound becomes an equality in the case where all entries of $C^\top C$ are non-negative.

---

[2]For simplicity of exposition, we use an implicit learning rate of 1. Because of the linearity of the operations, the general case can be recovered by pre-scaling $x_1, \ldots, x_n$ accordingly.

**BISR noise correlation (p = 3)**

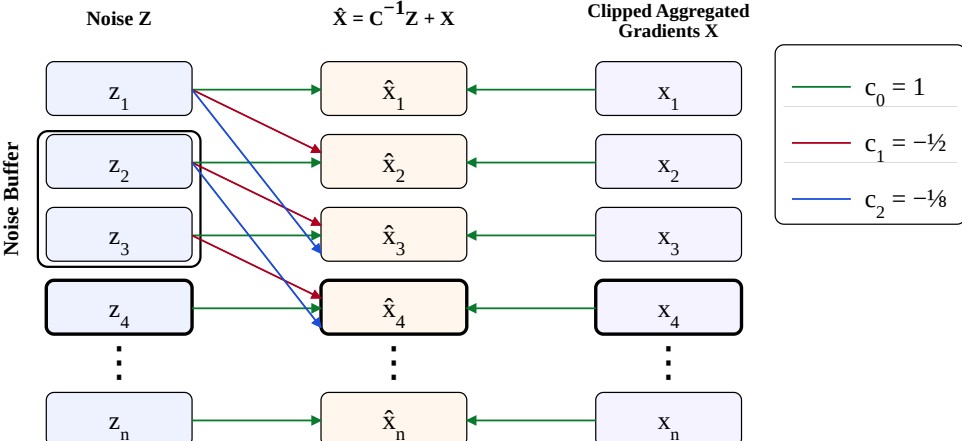

Figure 1: Illustration of noise correlation using the Banded Inverse Square Root (BISR) method with bandwidth $p = 3$. For example, at step 4, we form a new aggregate from a sampled batch of clipped gradients $x_4$. We take the noise vectors $z_2$ and $z_3$ from the Noise Buffer and add them to the gradient with coefficients $c_1 = -\frac{1}{2}$ and $c_2 = -\frac{1}{8}$, respectively. We then generate a new noise vector $z_4$ and add it directly to obtain the private estimate $\hat{x}_4$. Finally, we remove $z_2$ from the buffer and insert $z_4$ in its place.

## 2.2 Privacy Guarantees

The privacy analysis of our mechanism relies on the well-established Gaussian mechanism, which allows achieving arbitrary $(\varepsilon, \delta)$-differential privacy levels by calibrating the noise variance to the sensitivity of the underlying query. In particular, Theorem 2.1 of Denisov et al. (2022) states that the guarantees proved for the nonadaptive continual release model extend to the fully adaptive setting under a suitable factorization of the query matrix. This ensures that the same differential privacy guarantees hold even when the input stream is chosen adaptively. A related multi-epoch variant was studied in Choquette-Choo et al. (2023a), which corresponds to a specific choice of the notion of neighboring streams.

**Theorem 1.** *(Denisov et al., 2022, Theorem 2.1) Let $A \in \mathbb{R}^{n \times n}$ be a lower-triangular full-rank query matrix, and let $A = BC$ be any factorization with the following property: for any two neighboring streams of vectors $G, H \in \mathbb{R}^{n \times d}$, differing only in positions of participation of a single user, we have $\|C(G - H)\|_F \leq \kappa$. Let $Z \sim \mathcal{N}(0, \kappa^2 \sigma^2)^{n \times d}$ with $\sigma$ large enough so that*

$$\mathcal{M}(G) \;=\; AG + BZ \;=\; B(CG + Z)$$

*satisfies $(\varepsilon, \delta)$-DP in the nonadaptive continual release model. Then $\mathcal{M}$ satisfies the same DP guarantee (with the same parameters) even when the rows of the input are chosen adaptively.*

Formally, the privacy guarantee of the Gaussian mechanism itself is characterized by the following analytic condition due to Balle & Wang (2018), which provides the exact relationship between the sensitivity, the privacy parameters, and the variance of the added Gaussian noise:

**Theorem 2** *(Analytic Gaussian Mechanism (Balle & Wang, 2018)). Let $f : \mathcal{X} \to \mathbb{R}^d$ be a function with global $\ell_2$ sensitivity $\Delta$. For any $\varepsilon \geq 0$ and $\delta \in [0, 1]$, the Gaussian output perturbation mechanism $M(x) = f(x) + Z$ with $Z \sim \mathcal{N}(0, \sigma^2 \mathbb{I}_d)$ is $(\varepsilon, \delta)$-DP if and only if*

$$\Phi\left(\frac{\Delta}{2\sigma} - \frac{\varepsilon\sigma}{\Delta}\right) - e^\varepsilon \, \Phi\left(-\frac{\Delta}{2\sigma} - \frac{\varepsilon\sigma}{\Delta}\right) \;\leq\; \delta.$$

This condition enables tight calibration of the noise level $\sigma$ for any target privacy parameters $(\varepsilon, \delta)$.

**Optimal factorization.** Better choices of factorization matrices can achieve the same privacy levels with less added noise, potentially leading to higher utility. Therefore, various factorizations have been proposed and studied theoretically as well as empirically.

---

**Algorithm 1** Differentially Private SGD with Banded Inverse Matrix Factorization

---

**Input:** Initialization $\theta_0 \in \mathbb{R}^d$, dataset $\mathcal{D}$, batch size $B$, clip norm $\zeta$, learning rate $\eta > 0$, weight decay $0 < \alpha \leq 1$, momentum $0 \leq \beta < 1$, loss $\ell(\theta, d)$, noise multiplier $\sigma_{\epsilon, \delta} > 0$, coefficients of the banded inverse Toeplitz correlation matrix $C^{-1}$: $(c_0, \ldots, c_{p-1})$

1: Initialize $m_0 \leftarrow \mathbf{0} \in \mathbb{R}^d$
2: **for** $i = 1$ to $n$ **do**
3:      Sample a minibatch $S_i = \{d_1, \ldots, d_B\} \subseteq \mathcal{D}$
4:      **for** $j = 1$ to $B$ **do**
5:          $g_j \leftarrow \nabla_\theta \, \ell(\theta_{i-1}, d_j)$
6:          $\tilde{g}_j \leftarrow \min\left(1, \frac{\zeta}{\|g_j\|}\right) g_j$                                          ▷ per-example clipping
7:      $x_i \leftarrow \sum_{j=1}^{B} \tilde{g}_j$                                            ▷ aggregate clipped gradients
8:      Draw $Z_i \sim \mathcal{N}(0, \sigma_{\epsilon, \delta}^2 \mathbb{I}_d)$
9:      $\hat{x}_i \leftarrow x_i + \zeta \sum_{t=0}^{\min(p, i)-1} c_t \, Z_{i-t}$                      ▷ BISR noise injection
10:      $m_i \leftarrow \beta \, m_{i-1} + \hat{x}_i$                                  ▷ momentum update
11:      $\theta_i \leftarrow \alpha \, \theta_{i-1} - \eta \, m_i$                                 ▷ weight decay + step
12: **return** $\theta_n$

---

Choquette-Choo et al. (2023a) defines the *optimal factorization* as the one that minimizes the expected approximation error (1), and proposed an optimization problem to (approximately) compute this factorization. A downside of this approach is that the optimization problem is computationally expensive and the numeric solution does not provide theoretical insights, such as the optimal (i.e. lowest) rate of growth of the approximation error.

On the other hand, a square root factorization introduced by Henzinger et al. (2024), is an explicit factorization, defined by $A_{\alpha, \beta} = C_{\alpha, \beta}^2$, with positive main diagonal. Kalinin & Lampert (2024) showed that the factorization error of the square root factorization under multi-epoch participation is worse than that of the optimal factorization and they introduced *banded square root* (BSR) factorization, which is defined by making the matrix $C_{\alpha, \beta}$ banded. A limitation of BSR is that its guarantees are implicit in terms of the used bandwidth, which does not allow concluding how they relate to the optimal multi-epoch factorization at a theoretical level.

## 3    BANDED INVERSE SQUARE ROOT FACTORIZATION

In this section, we present our main theoretical results: we prove a new lower bound on the achievable approximation error (Theorem 3), we introduce the BISR factorization (Definition 1), and we prove that BISR achieves this (therefore optimal) rate (Theorem 4).

We first show an improved version of the lower bounds of the approximation error for general factorizations from Kalinin & Lampert (2024).

**Theorem 3** (General Multi-Participation Lower Bound). *Let $A_{\alpha, \beta} \in \mathbb{R}^{n \times n}$ be the SGD workload matrix (4), with momentum $\beta > 0$ and weight decay $\alpha > 0$. In the multi-participation setting with separation $1 \leq b \leq n$ and $k = \lceil \frac{n}{b} \rceil$, for any factorization $A_{\alpha, \beta} = BC$, it holds*

$$\mathcal{E}(B, C) = \begin{cases} \Omega(\sqrt{k} \log n + k) & \text{if} \quad \alpha = 1, \\ \Omega_\alpha(\sqrt{k}) & \text{if} \quad \alpha < 1. \end{cases} \tag{6}$$

*Proof Sketch.* We use the probabilistic method in Lemma 6 to obtain the bounds $\Omega_\alpha(\sqrt{k})$ for $\alpha < 1$ and $\mathcal{E}(B, C) = \Omega(\sqrt{k} \log n)$ for $\alpha = 1$. It remains to prove that for $\alpha = 1$, we also have the lower bound $\mathcal{E}(B, C) = \Omega(k)$. For that, we prove a general inequality: given a valid participation vector $\boldsymbol{\pi}$, we have $\mathcal{E}(B, C) \geq \frac{1}{n} \|A_{1, \beta} \boldsymbol{\pi}\|_2$. $\qquad\square$

As our second main contribution, we now introduce the BISR factorization for multi-epoch SGD.

**Definition 1** (Banded Inverse Square Root (BISR)). *For a given workload matrix $A$, let $C$ be the matrix square root (i.e. $C^2 = A$) with positive values on the diagonal. Let $C^p$ be the matrix obtained*

*by: i) computing the inverse matrix $C^{-1}$, ii) imposing a banded structure with $p$ bands by setting all elements below the $p$-th diagonal to zero, iii) inverting the resulting banded matrix back. Then, we denote by BISR the matrix factorization $A = B^p C^p$, with $B^p = A(C^p)^{-1}$.*

BISR can be seen as an alternative realization of the insights behind the BSR (Banded Square Root) factorization from Kalinin & Lampert (2024). There, the intuition was that making the matrix $C$ $p$-banded reduces its sensitivity without increasing the Frobenius norm of the subsequent postprocessing matrix too much, thereby resulting in an overall reduction of the approximation error. The authors did not derive exact rates, though, because the dependence on $p$ is not explicit.

For BISR, we instead make the matrix $C^{-1}$ $p$-banded. This also leads to a reduction of the approximation error compared to the non-banded case, but with two additional advantages. First, the resulting algorithm (see Algorithm 1 and Figure 1 for the illustration) is time- and memory-efficient because the product of $(C^p)^{-1} Z$ can be represented as a convolution with $p$ elements. Therefore, the computation can be performed efficiently: in a streaming setting, only $p$ rows of the matrix $Z$ need to be stored at any time, while in an offline setting (which requires more storage), it can be accelerated further using the Fast Fourier Transform. Second, and mainly, it allows us to derive more explicit expressions of the approximation error with respect to the bandwidth $p$. In particular, we show the following.

**Theorem 4** (BISR Approximation Error). *For $1 \le p \le n$ and $1 \le k \le \frac{n}{b}$ the following upper bound holds for the matrix factorization error of the BISR $A_{\alpha,\beta} = B^p_{\alpha,\beta} C^p_{\alpha,\beta}$ (as in Definition 1):*

$$\mathcal{E}(B^p_{\alpha,\beta}, C^p_{\alpha,\beta}) = \begin{cases} O_\beta \left( \sqrt{k} \log p + \sqrt{\frac{nk}{b}} + \sqrt{\frac{nk \log p}{p}} + \sqrt{\frac{kp \log p}{b}} \right) & for \quad \alpha = 1, \\ O_{\alpha,\beta}(\sqrt{k}) & for \quad \alpha < 1, \end{cases} \tag{7}$$

*where $O_{\alpha,\beta}$ and $O_\beta$ indicate that the hidden constant may depend on $\alpha$ and $\beta$, respectively.*

*Proof sketch.* To prove Theorem 4, we use Lemma 11 to bound the Frobenius norm $\|B^p_{\alpha,\beta}\|_F$. Then, Theorem 2 from Kalinin & Lampert (2024) (see Theorem 5), together with monotonicity of values $C^p_{\alpha,\beta}$ from Lemma 8, provides an explicit way to express the sensitivity $\mathrm{sens}_{k,b}(C_{\alpha,\beta})$. To bound the sensitivity, we apply Lemma 12 to bound individual values. The product of the bounds on $\|B^p_{\alpha,\beta}\|_F$ and $\mathrm{sens}_{k,b}(C_{\alpha,\beta})$ yields the result. The full proof of Theorem 4 can be found in the appendix. □

For comparison, Kalinin & Lampert (2024) proved a bound $O\left( \sqrt{\frac{nk \log p}{p}} \right) + O_p(\sqrt{k})$ on the approximation error of the BSR in the case $\alpha = 1$, $\beta = 0$. While the first term also appears in (7), the second is non-informative about the effect of the bandwidth, $p$. Therefore, it does not allow making a statement about the optimality of the BSR.

In contrast, Theorem 4 is explicit about the role of $p$. Choosing its value to be $O(b \log b)$, such that the occurring terms in (7) are minimized, we obtain the following corollary.

**Corollary 1** (Optimized BISR Approximation Error). *Let $A_{\alpha,\beta} = B^p_{\alpha,\beta} C^p_{\alpha,\beta}$ be the BISR factorization defined of Definition 1. For $1 \le b \le n$ let $k = \lceil \frac{n}{b} \rceil$. Then, for $p^* = O(b \log b)$, the matrix factorization error admits the following optimized upper bound:*

$$\mathcal{E}(B^{p^*}_{\alpha,\beta}, C^{p^*}_{\alpha,\beta}) = \begin{cases} O_\beta \left( \sqrt{k} \log n + k \right), & for \, \alpha = 1, \\ O_{\alpha,\beta}(\sqrt{k}), & for \, \alpha < 1. \end{cases} \tag{8}$$

Comparing the upper bound in eq. (8) with the lower bound in eq. (6), we obtain that BISR is an **asymptotically optimal** factorization in the multi-participation setting.

To use the BISR factorization, we apply the following lemma, which provides analytic expressions for the elements of the inverse matrix $C^{-1}_{\alpha,\beta}$.

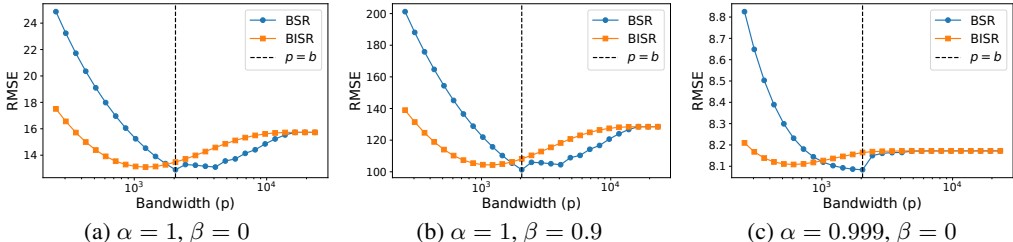

Figure 2: RMSE comparison for Banded Square Root (BSR) and Banded Inverse Square Root (BISR) methods across varying bandwidths ($p$). The results are shown for a fixed matrix size of $n = 16384$ and a participation number of $k = 8$. For BSR, the choice $p = b$ is numerically optimal, while for BISR a smaller bandwidth suffices to achieve an optimal value.

**Lemma 1** (Inverse Square Root of the Matrix $A_{\alpha,\beta}$). *For $k \geq 0$, let $\tilde{r}_k = (-1)^k \binom{1/2}{k} = \frac{-r_k}{2k-1} = \frac{-1}{2k-1} \left| \binom{-1/2}{k} \right|$. The inverse square root of the matrix $A_{\alpha,\beta}^{-1/2}$ defined in* (4), *for $0 \leq \beta < \alpha \leq 1$, is*

$$C_{\alpha,\beta}^{-1} = \begin{pmatrix} 1 & 0 & \dots & 0 \\ \tilde{c}_1^{\alpha,\beta} & 1 & \dots & 0 \\ \vdots & \vdots & \ddots & \vdots \\ \tilde{c}_{n-1}^{\alpha,\beta} & \tilde{c}_{n-2}^{\alpha,\beta} & \dots & 1 \end{pmatrix}, \quad where \quad \tilde{c}_k^{\alpha,\beta} = \sum_{j=0}^{k} \tilde{r}_j \beta^j \tilde{r}_{k-j} \alpha^{k-j}. \tag{9}$$

The values of the matrix $C_{\alpha,\beta}^{-1}$ can then be computed efficiently via the Fast Fourier Transform (FFT) as a convolution of the sequences $\tilde{r}_j \alpha^j$ and $\tilde{r}_j \beta^j$, where the sequence $\tilde{r}_j$ for $j = 1, \dots, n$ can be computed in linear time using the recursive expression $\tilde{r}_j = \frac{j-3/2}{j} \tilde{r}_{j-1}$.

Following the work of Andersson & Yehudayoff (2025), we show that the space complexity of the matrix $(C_{\alpha,\beta}^p)^{-1}$ is equal to $p$, meaning that exact multiplication with a random real vector $z$ in a streaming setting, performing continuous operations, requires storing $p$ real values (not including the memory needed to store the matrix coefficients). This implies that, for memory-efficient computation, one must either use a small bandwidth $p$ or consider approximate multiplication. We formally state the result in the following lemma:

**Lemma 2.** *The space complexity–defined as the minimum buffer size required by a streaming algorithm to correctly process an input–for computing the product of the Toeplitz matrix $(C_{\alpha,\beta}^p)^{-1}$ with an arbitrary vector $z \in \mathbb{R}^n$, for $n \geq 2p - 1$, is exactly $p$, and at least $\frac{n-5}{2}$ for $C_{\alpha,\beta}^{-1}$.*

## 4 EXPERIMENTS

In this section, we present numerical results from evaluating various factorization methods in the multi-participation regime.

We first study the effect of using different bandwidths for BSR and BISR, as shown in Figure 2. We found that, in most settings, the optimal bandwidth for BSR coincides with the separation parameter $b$, whereas for BISR a smaller bandwidth suffices; therefore, in future comparisons with BISR, we propose optimizing the bandwidth to determine the optimal $p^*$.

We compare BISR with several other methods, including Banded Square Root (BSR), Banded Matrix Factorization (BandMF), introduced by McKenna (2025), and Buffered Linear Toeplitz (BLT), introduced by Dvijotham et al. (2024) and adapted for multi-participation training by McMahan et al. (2024). We use a buffer size of $4$, as recommended, and observe that the error saturates quickly as the buffer size increases. We use BandMF with bandwidth equal to $b$, as we did not observe any benefit from using a larger bandwidth. Moreover, we conjecture that optimal multi-epoch participation can always be achieved on a banded lower triangular matrix with bandwidth $b$.

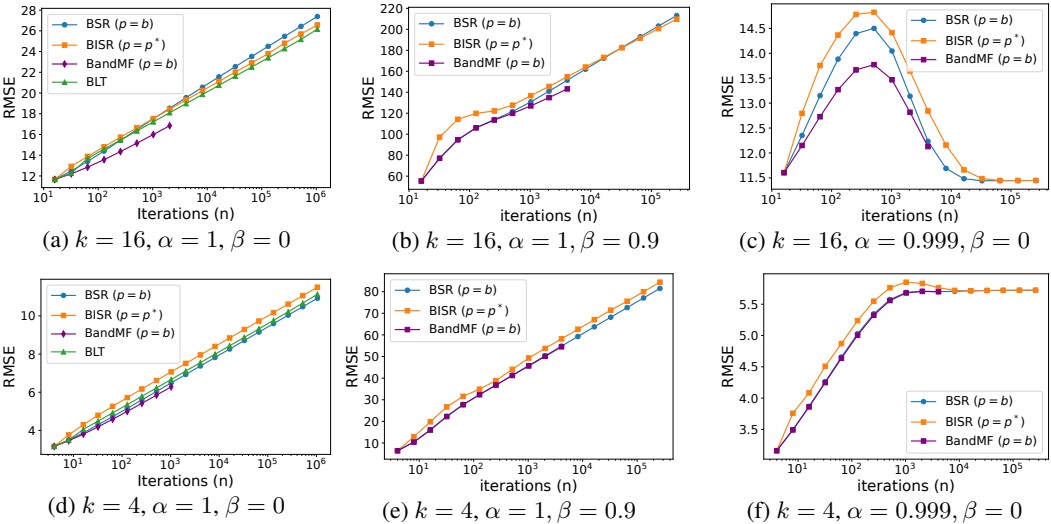

Figure 3: RMSE across varying matrix sizes for different factorization methods under multiple optimizer settings and participation levels. We showed that also in practice, BISR performs on par with, or even better than, BSR and BLT. However, methods based on numerical optimization, such as BandMF, may achieve superior performance in certain regimes.

We emphasize that BLT has only been described, analyzed, and implemented for prefix-sum matrices.[3] Therefore, we do not show BLT results for momentum and weight decay plots. For all methods except BISR, we use efficient implementations from the jax-privacy library (Balle et al., 2025).

Our experiments (Figure 3) show that banded inverse square root factorization consistently matches or exceeds BSR in quality across all regimes and outperforms it in scenarios with a large number of participations. The improvement is particularly pronounced when the participation count is high ($k = 16$). BISR achieves RMSE comparable to that of BLT, but has the advantage of easier implementation for both factorization and training, as it only requires convolving previous noise with a fixed set of coefficientsan "embarrassingly parallel" operation (see McKenna (2025)). While in practice, BandMF achieves slightly better RMSE[4] at $k = 16$, it requires solving a computationally expensive optimization problem, making it impractical for matrix sizes beyond $n = 4096$.

## 5 FROM BISR TO BANDINVMF

In the previous sections, we established that BISR has asymptotically optimal rate for large bandwidths $p \sim b \log b$. However, in practice, one might want to work with a smaller value of $p$ to save memory and computational resources. In this section, we showcase a modification to BISR with improved practical properties in this regime. We propose to keep the construction of a banded inverse matrix with Toeplitz structure, but to set its values not by the closed form expressions (9) but by a numeric optimization. Specifically, we optimize an upper bound on $b$-min separation participation, given by Equation 5.

For the sake of numerical optimization, we assume that the optimum is achieved for indices of the form $i + kb$. This assumption can be justified, as we observed that the resulting solution for matrix $C$ is positive and decreasing, which guarantees optimality. If the sequence is not decreasing, it can be uniformly bounded by a decreasing sequence. Specifically, if the values on the diagonals of $C$ are $C_{j,1}$, then they are upper bounded by $\max_{t \geq j} C_{1,t}$, which is a decreasing sequence by construction. We use banded inverse square root factorization as an initialization for the coefficients. We provide an efficient JAX implementation in the Appendix (see Algorithm 1) as well as the convergence plots in Figure 6.

---

[3]In concurrent work by Huang et al. (2025), BLT has been adapted to work with a momentum workload.

[4]This is possible because even though we have shown that BISR provides an asymptotically optimal factorization, that does not imply that it is superior to all other methods for finite problem sizes.

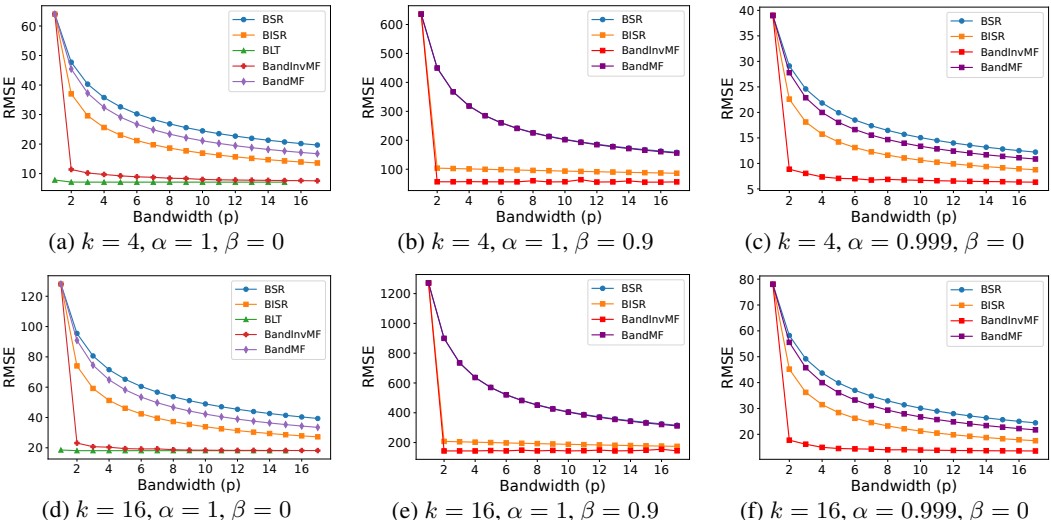

Figure 4: RMSE across different factorizations and optimization parameters $\alpha, \beta$, with small bandwidth. The plot shows that BandInvMF and BISR can significantly reduce RMSE for a small bandwidth regime, justifying the use of banded inverse methods instead of banded factorizations.

The numerical results are presented in Figure 4 referred to as BandInvMF. We observe that the error decreases drastically even with the addition of a single band, compared to a trivial factorization. This observation is supported theoretically by the following lemma.

**Lemma 3** (Optimal Band Inversion Error). *Let the matrix $C_\lambda^{-1} = \text{LTT}(1, -\lambda, 0, \ldots, 0)$ be a lower triangular Toeplitz matrix with $1$ on the main diagonal and $-\lambda$ on the subdiagonal. Then, for a single participation and the prefix sum matrix $A_{1,0}$, the following bound on the matrix factorization error holds:*

$$\inf_{\lambda \in (0,1)} \mathcal{E}\big(A_{1,0} C_\lambda^{-1}, C_\lambda\big) = O(n^{1/4}). \tag{10}$$

Lemma 3 shows that the optimized inverse banded matrix factorization can achieve an asymptotically better bound than a trivial factorization $A \times \mathbb{I}$, which yields an error of $O(\sqrt{n})$. Moreover, from Theorem 4, for small $p$, the leading term for banded inverse square root factorization remains of order $O(\sqrt{n})$. Therefore, we advocate for optimizing the coefficients when the bandwidth is small.

We compare Band-Inv-MF with other methods for training the 3-Block ConvNet model on CIFAR-10 and the BERT-base model ($\sim$100M parameters) on IMDB (see Figure 5), both with and without amplification by subsampling. For a fairer comparison, we use a recently proposed bins-and-balls subsampling mechanism (Chua et al., 2025), which combines the accuracy benefits of Poisson subsampling with improved implementation efficiency. More importantly, it supports the matrix mechanism via the MCMC accountant (Choquette-Choo et al., 2024b;c), even when the matrix $C$ is not banded. Our results indicate that in a low-memory regime, inverse correlation matrix methods BISR and Band-Inv-MF achieve significantly higher accuracy than BSR and Band-MF, and consistently outperform DP-SGD, with and without amplification. Although Band-Inv-MF achieves lower RMSE than BISR, we did not observe a corresponding gain in accuracy. This indicates that RMSE alone is not a sufficient proxy for model performance in matrix factorization.

## 6 DISCUSSION

This work demonstrates that imposing a banded structure on the inverse correlation matrix, rather than on the matrix itself, leads to both theoretical and practical benefits for differentially private training across multiple participations. Our Banded Inverse Square Root (BISR) method enables explicit factorization, supporting clean error analysis and efficient implementation.

We prove that BISR achieves asymptotically optimal factorization error by improving upon previously established lower bounds and showing that BISR matches the asymptotics precisely, thereby closing a

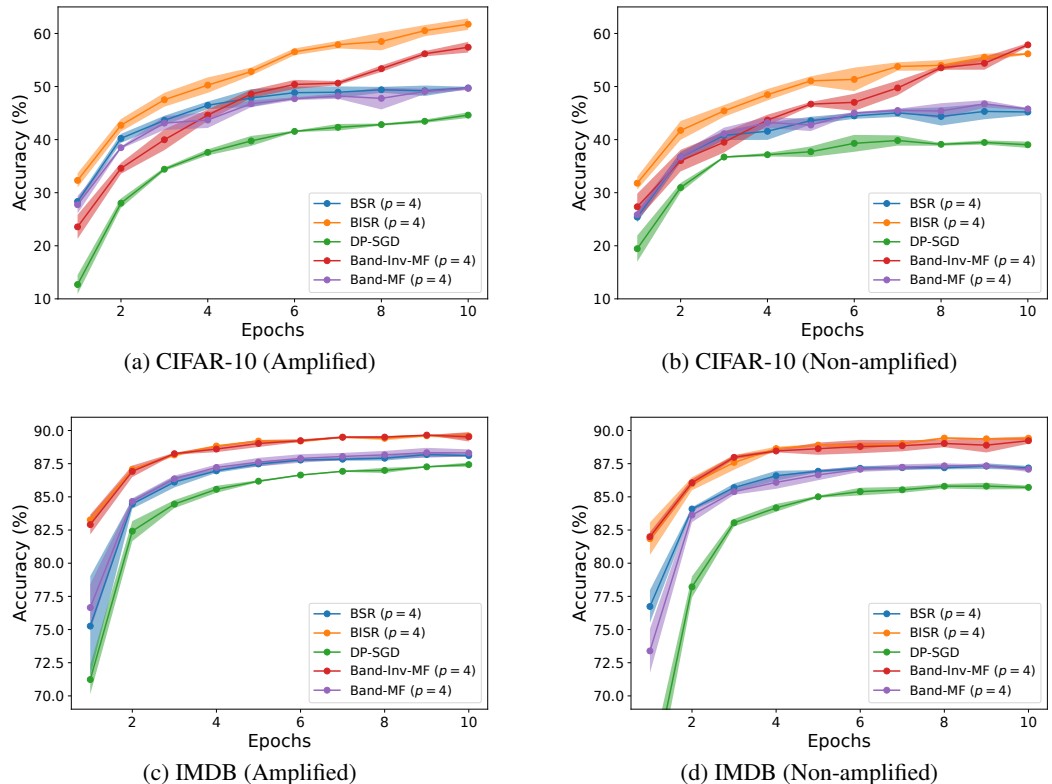

Figure 5: **Accuracy results for CIFAR-10 and IMDB in the small bandwidth (low-memory) regime.** For CIFAR-10, both amplified (left) and non-amplified (right) results show that inverse factorization methods, BISR and Band-Inv-MF, achieve significantly higher accuracy compared to Band-MF. Both plots correspond to $(9, 10^{-5})$-DP, with training performed using momentum $\beta = 0.9$ and weight decay $\alpha = 0.9999$, which we found to be optimal (see Tables 1 and 2 in the appendix). For IMDB, we report accuracy from fine-tuning under the same low-memory regime, comparing amplified and non-amplified settings, with training performed using momentum $\beta = 0.95$ and weight decay $\alpha = 0.99999$ (see Tables 3 and 4).

fundamental gap in the theory. An interesting direction for future work is to close the gap in constant dependence, as numerical optimization methods (e.g., BandMF, BLT), despite their computational cost, may outperform BISR.

In the low-memory regime, we find it beneficial to optimize directly over the coefficients of the inverse correlation matrix. Our Band-Inv-MF method achieves a lower matrix factorization error compared to BISR. However, these improvements do not yet translate to gains in model accuracy when training with the amplification by subsampling. Future research should focus on optimizing the matrix coefficients while explicitly accounting for amplification, in order to bridge this gap.

## ACKNOWLEDGMENTS

We thank Arun Ganesh for providing the code for the MCMC accountant. We thank Joel Andersson, Rasmus Pagh and Monika Henzinger for valuable comments on the early version of the draft. We thank Christian Lebeda for a fruitful discussion on the lower bound theorem.

Jalaj Upadhyays research was funded by the Rutgers Decanal Grant no. 302918, NSF CNS 2433628, Google Research Scholar Award, and Google Seed Fund Grant.

Nikita Kalinin's research was funded in part by the Austrian Science Fund (FWF) [10.55776/COE12].

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

## KEY INEQUALITIES AND RELATIONSHIPS

This section compiles the fundamental inequalities and key relationships employed throughout this study. Each entry is presented with a concise explanation of its origin or the context in which it arises.

1. $r_k = \left| \binom{-1/2}{k} \right| = \frac{1}{4^k} \binom{2k}{k}$  *Henzinger et al. (2024)*

2. $\dfrac{1}{\sqrt{\pi(k+1)}} \leq r_k \leq \dfrac{1}{\sqrt{\pi k}}$  *Lemma 2.1 from Dvijotham et al. (2024)*

3. $\displaystyle\sum_{k=0}^{p-1} r_k \leq 1 + \frac{1}{\sqrt{\pi}} \sum_{k=1}^{p-1} \frac{1}{\sqrt{k}} \leq 1 + \frac{2\sqrt{p}}{\sqrt{\pi}}$  *Integral inequality.*

4. $\displaystyle\sum_{k=0}^{p-1} r_k^2 \leq 1 + \log p$  *Lemma 5*

5. $c_k^{\alpha,\beta} = \displaystyle\sum_{j=0}^{k} \alpha^j \beta^{k-j} r_j r_{k-j}$  *Theorem 1 from Kalinin & Lampert (2024)*

6. $c_k^{1,\beta} \leq c_{k-1}^{1,\beta} \left[ 1 - \frac{(1-\beta)^2}{2k} \right] \quad$ for $k \geq 1$.  *Lemma 10*

7. $\displaystyle\sum_{j=0}^{k} c_j^{1,\beta} = \sum_{j=0}^{k} r_j \beta^j \tilde{r}_{k-j}$  *In the proof of Lemma 9*

8. $\displaystyle\sum_{j=0}^{k} c_j^{1,\beta} \beta^{k-j} = \sum_{j=0}^{k} r_j^\beta \tilde{r}_{k-j}$  *In the proof of Lemma 9*

9. $\displaystyle\sum_{j=0}^{k} \frac{\tilde{c}_j^{1,\beta}(1-\beta^{k-j+1})}{1-\beta} = c_k^{1,\beta}$  *In the proof of Lemma 9*

10. $r_k(1-\beta) \leq \displaystyle\sum_{j=0}^{k} \tilde{c}_j^{1,\beta} \leq c_k^{1,\beta}(1-\beta)$  *Lemma 9*

11. $\dfrac{\log(k+1)}{4} \leq \displaystyle\sum_{j=0}^{k-1} (c_j^{1,\beta})^2 \leq \frac{1+\log k}{(1-\beta)^2}$  *Lemma 5*

12. $\dfrac{\alpha^j}{2\sqrt{j+1}} \leq c_j^{\alpha,\beta} \leq \dfrac{\alpha^j}{(1-\beta/\alpha)\sqrt{j+1}}$  *Lemma 5*

13. $1 \leq \displaystyle\sum_{j=0}^{k-1} (c_j^{\alpha,\beta})^2 \leq \frac{1}{(\alpha-\beta)^2} \log\left(\frac{1}{1-\alpha^2}\right)$  *Lemma 5*

14. $\tilde{r}_k = (-1)^k \binom{1/2}{k} = \dfrac{-1}{2k-1} r_k$  *Lemma 1*

15. $\tilde{c}_k^{\alpha,\beta} = \displaystyle\sum_{j=0}^{k} \tilde{r}_j \beta^j \tilde{r}_{k-j} \alpha^{k-j}$  *Lemma 1*

16. $\tilde{r}_k(1+\beta) \leq \tilde{c}_k^{1,\beta} \leq 0$  *Lemma 7*

## A    PROOFS

We start by stating some auxiliary results that will be extensively used throughout the proofs.

**Lemma 4** (Theorem 1 in Kalinin & Lampert (2024) – Square Root of the Matrix $A_{\alpha,\beta}$). *For $k \geq 0$, let $r_k = \left| \binom{-1/2}{k} \right|$. For $0 \leq \beta < \alpha \leq 1$, the square root matrix $C_{\alpha,\beta} = A_{\alpha,\beta}^{1/2}$ has the following explicit form:*

$$C_{\alpha,\beta} = \begin{pmatrix} 1 & 0 & \cdots & 0 \\ c_1^{\alpha,\beta} & 1 & \cdots & 0 \\ \vdots & \vdots & \ddots & \vdots \\ c_{n-1}^{\alpha,\beta} & c_{n-2}^{\alpha,\beta} & \cdots & 1 \end{pmatrix}, \quad \text{where} \quad c_k^{\alpha,\beta} = \sum_{j=0}^{k} \alpha^j \beta^{k-j} r_j r_{k-j}, \tag{11}$$

**Theorem 5** (Theorem 2 from Kalinin & Lampert (2024)). *Let $M$ be a lower triangular Toeplitz matrix with decreasing non-negative entries $m_0 \geq m_1 \geq m_2 \geq \ldots m_{n-1} \geq 0$ on the diagonals. Then the* sensitivity *of $M$ in the setting of $b$-min-separation is*

$$\text{sens}_{k,b}(M) = \left\| \sum_{j=0}^{k-1} M_{[\cdot, 1+jb]} \right\|_2 \tag{12}$$

*where $M_{[\cdot, 1+jb]}$ denotes the $(1+jb)$-th column of $M$.*

**Lemma 5** (Lemma 7 from Kalinin & Lampert (2024)). *For $k \in \{1, \ldots, n\}$ it holds for $c_i^{\alpha,\beta}$ as defined in equation (11):*

$$\frac{\log(k+1)}{4} \leq \sum_{i=0}^{k-1} (c_i^{1,\beta})^2 \leq \frac{1 + \log k}{(1-\beta)^2} \tag{13}$$

*for $\alpha = 1$, and otherwise*

$$1 \leq \sum_{i=0}^{k-1} (c_i^{\alpha,\beta})^2 \leq \frac{1}{(\alpha - \beta)^2} \log\left(\frac{1}{1-\alpha^2}\right). \tag{14}$$

**Lemma 2.** *The space complexity–defined as the minimum buffer size required by a streaming algorithm to correctly process an input–for computing the product of the Toeplitz matrix $(C_{\alpha,\beta}^p)^{-1}$ with an arbitrary vector $z \in \mathbb{R}^n$, for $n \geq 2p - 1$, is exactly $p$, and at least $\frac{n-5}{2}$ for $C_{\alpha,\beta}^{-1}$.*

*Proof of Lemma 2.* Throughout this proof, we use only results from Andersson & Yehudayoff (2025), and thus any reference to a theorem or lemma should be understood as coming from that work. We use their Lemma 7, which lower-bounds the space complexity of a Toeplitz matrix using the rank of a corresponding Hankel matrix of its coefficients. The matrix $C_{\alpha,\beta}^{-1}$ has a generating function $f = \sqrt{(1 - \alpha x)(1 - \beta x)}$. The proof of their Corollary 16 implies that the Hankel matrix $H[f]$ has corank at most 3. Thus, Lemma 7 implies that $C_{\alpha,\beta}^{-1}$ has space complexity at least $\frac{n+1}{2} - 3$. For the matrix $(C_{\alpha,\beta}^p)^{-1}$, the generating function is a rational function of degree $p$; therefore, for $n \geq 2p - 1$, their Theorem 2 implies that the space complexity is exactly $p$, concluding the proof.    $\square$

**Theorem 3** (General Multi-Participation Lower Bound). *Let $A_{\alpha,\beta} \in \mathbb{R}^{n \times n}$ be the SGD workload matrix (4), with momentum $\beta > 0$ and weight decay $\alpha > 0$. In the multi-participation setting with separation $1 \leq b \leq n$ and $k = \lceil \frac{n}{b} \rceil$, for any factorization $A_{\alpha,\beta} = BC$, it holds*

$$\mathcal{E}(B,C) = \begin{cases} \Omega(\sqrt{k}\log n + k) & \text{if} \quad \alpha = 1, \\ \Omega_\alpha(\sqrt{k}) & \text{if} \quad \alpha < 1. \end{cases} \tag{6}$$

*Proof of Theorem 3.* We use the probabilistic method in Lemma 6 to obtain the bounds $\Omega_\alpha(\sqrt{k})$ for $\alpha < 1$ and $\mathcal{E}(B,C) = \Omega(\sqrt{k}\log n)$ for $\alpha = 1$. It remains to prove that for $\alpha = 1$, we also have the lower bound $\mathcal{E}(B,C) = \Omega(k)$.

We begin with the following observation: given a matrix $C$, we can compute an optimal participation scheme represented by a vector with ones at positions corresponding to participating columns,

denoted by $\pi_C^*$. As a lower bound, we consider a specific participation vector $\pi_1$, with ones in columns indexed by $1 + ib$ for $i \in [0, k-1]$, such that $|\pi_1| = k$. By construction, we have $\text{sens}_{k,b}(C) := \|C\pi_C^*\|_2 \geq \|C\pi_1\|_2$. Therefore, the error can be bounded as follows:

$$\mathcal{E}(B, C) = \frac{1}{\sqrt{n}}\|B\|_F \, \text{sens}_{k,b}(C) \geq \frac{1}{\sqrt{n}}\|B\|_F\|C\pi_1\|_2 \geq \frac{1}{\sqrt{n}}\|BC\pi_1\|_2 = \frac{1}{\sqrt{n}}\|A_{1,\beta}\pi_1\|_2. \quad (15)$$

As a lower bound, we consider $\beta = 0$ as $\|A_{1,\beta}\pi_1\|_2 \geq \|A_{1,0}\pi_1\|_2$. The elements of the matrix $A_{1,0}$ are positive and non-increasing. Therefore, by Theorem 5, the $(k, b)$-sensitivity of $A_{1,0}$ is exactly $\|A_{1,0}\pi_1\|_2$. By Theorem 9 from Kalinin & Lampert (2024), this sensitivity is at least $\frac{k\sqrt{n}}{\sqrt{3}}$, resulting in the lower bound:

$$\mathcal{E}(B, C) \geq \frac{k}{\sqrt{3}} = \Omega(k), \quad (16)$$

which concludes the proof. $\qquad\square$

**Lemma 6.** *Let $A_{\alpha,\beta} \in \mathbb{R}^{n \times n}$ be the SGD workload matrix (4). In the multi-participation setting with separation $1 \leq b \leq n$ and $k = \lceil \frac{n}{b} \rceil$, for any factorization $A_{\alpha,\beta} = BC$, it holds that*

$$\mathcal{E}(B, C) = \begin{cases} \Omega(\sqrt{k}\log n), & \alpha = 1 \\ \Omega(\sqrt{k}), & \alpha < 1 \end{cases} \quad (17)$$

*Proof.* Here, we refine Theorem 8 from Kalinin & Lampert (2024) by removing the assumption that the scalar products between the columns of the matrix $C$ are non-negative, i.e., $C^\top C \geq 0$. We first prove that

$$\text{sens}_{k,b}(C)^2 \geq \frac{1}{4b}\|C\|_F^2. \quad (18)$$

To do so, we lower bound the $b$-min separation participation by the $(k, b)$-participation, where we have a fixed $b$ separation between vectors but are allowed to include only a subset of them. This splits the set of all column indices into $b$ disjoint subsets $\mathcal{S}_j$ for $j \in [1, b]$ with $|S_j| \leq k$. Then, the following inequality holds:

$$\text{sens}_{k,b}(C)^2 \geq \max_{j \in [1,b]} \sup_{S \subseteq \mathcal{S}_j} \left\| \sum_{i \in S} C_{[:,i]} \right\|_2^2, \quad (19)$$

where $C_{[:,i]}$ denotes the $i$-th column of the matrix $C$.

To prove a lower bound, we use the probabilistic method. Consider i.i.d. random variables $\epsilon_i \sim \text{Bernoulli}(\frac{1}{2})$. Then:

$$\sup_{S \subseteq \mathcal{S}_j} \left\| \sum_{i \in S} C_{[:,i]} \right\|_2^2 \geq \mathbb{E}\left\| \sum_{i \in \mathcal{S}_j} C_{[:,i]}\epsilon_i \right\|_2^2 = \frac{1}{2} \sum_{i \in \mathcal{S}_j} \|C_{[:,i]}\|_2^2 + \frac{1}{4} \sum_{\substack{i \neq i' \\ i,i' \in \mathcal{S}_j}} \langle C_{[:,i]}, C_{[:,i']} \rangle$$

$$= \frac{1}{4} \sum_{i \in \mathcal{S}_j} \|C_{[:,i]}\|_2^2 + \frac{1}{4}\left\| \sum_{i \in \mathcal{S}_j} C_{[:,i]} \right\|_2^2 \geq \frac{1}{4} \sum_{i \in \mathcal{S}_j} \|C_{[:,i]}\|_2^2. \quad (20)$$

Thus,

$$\text{sens}_{k,b}(C)^2 \geq \max_{j \in [1,b]} \frac{1}{4} \sum_{i \in \mathcal{S}_j} \|C_{[:,i]}\|_2^2 \geq \frac{1}{4b} \sum_{i=1}^{n} \|C_{[:,i]}\|_2^2 = \frac{1}{4b}\|C\|_F^2. \quad (21)$$

Therefore,

$$\mathcal{E}(B, C) = \frac{1}{\sqrt{n}}\|B\|_F \, \text{sens}_{k,b}(C) \geq \frac{1}{2\sqrt{nb}}\|B\|_F\|C\|_F \geq \frac{1}{2\sqrt{nb}}\|BC\|_* = \frac{1}{2\sqrt{nb}}\|A_{\alpha,\beta}\|_*. \quad (22)$$

The nuclear norm of the matrix $A_{\alpha,\beta}$ has been lower bounded in Lemma 8 of Kalinin & Lampert (2024) by $\Omega(n \log n)$ for $\alpha = 1$, and by $\Omega(n)$ for $\alpha < 1$. Substituting $k = \lceil \frac{n}{b} \rceil$ concludes the proof. $\qquad\square$

**Lemma 1** (Inverse Square Root of the Matrix $A_{\alpha,\beta}$). *For $k \geq 0$, let $\tilde{r}_k = (-1)^k \binom{1/2}{k} = \frac{-r_k}{2k-1} = \frac{-1}{2k-1} \left| \binom{-1/2}{k} \right|$. The inverse square root of the matrix $A_{\alpha,\beta}^{-1/2}$ defined in (4), for $0 \leq \beta < \alpha \leq 1$, is*

$$
C_{\alpha,\beta}^{-1} = \begin{pmatrix} 1 & 0 & \cdots & 0 \\ \tilde{c}_1^{\alpha,\beta} & 1 & \cdots & 0 \\ \vdots & \vdots & \ddots & \vdots \\ \tilde{c}_{n-1}^{\alpha,\beta} & \tilde{c}_{n-2}^{\alpha,\beta} & \cdots & 1 \end{pmatrix}, \quad \text{where} \quad \tilde{c}_k^{\alpha,\beta} = \sum_{j=0}^{k} \tilde{r}_j \beta^j \tilde{r}_{k-j} \alpha^{k-j}. \tag{9}
$$

*Proof.* The matrix for the momentum matrix is given by:

$$
A_{\alpha,\beta} = A_{\alpha,0} \times A_{\beta,0} = \begin{pmatrix} 1 & 0 & \cdots & 0 \\ \alpha & 1 & \cdots & 0 \\ \vdots & \vdots & \ddots & \vdots \\ \alpha^{n-1} & \alpha^{n-2} & \cdots & 1 \end{pmatrix} \times \begin{pmatrix} 1 & 0 & \cdots & 0 \\ \beta & 1 & \cdots & 0 \\ \vdots & \vdots & \ddots & \vdots \\ \beta^{n-1} & \beta^{n-2} & \cdots & 1 \end{pmatrix}. \tag{23}
$$

The inverse square root then takes the form:

$$
C_{\alpha,\beta}^{-1} = C_{\alpha,0}^{-1} \times C_{\beta,0}^{-1}, \tag{24}
$$

since all lower triangular Toeplitz (LTT) matrices commute (see Strang (1986) or Böttcher & Grudsky (2000)). Therefore, it suffices to prove that the inverse square root of the matrix $C_{\alpha,0}^{-1}$ is a lower triangular Toeplitz matrix with elements $\tilde{r}_i \alpha^i$, which would imply the stated formula for $\tilde{c}_k^{\alpha,\beta}$, since the product of LTT matrices is given by the convolution of their elements.

The proof for the matrix $C_{\alpha,0}^{-1}$ is based on the identities of the generating functions of the sequences $r_k$ and $\tilde{r}_k$, derived simultaneously using the binomial formula:

$$
\begin{aligned}
(1 - \alpha x)^{-1/2} &= \sum_{k=0}^{\infty} \binom{-1/2}{k} (-1)^k \alpha^k x^k = \sum_{k=0}^{\infty} r_k \alpha^k x^k, \\
(1 - \alpha x)^{1/2} &= \sum_{k=0}^{\infty} \binom{1/2}{k} (-1)^k \alpha^k x^k = \sum_{k=0}^{\infty} \tilde{r}_k \alpha^k x^k.
\end{aligned} \tag{25}
$$

Then the generating function of the product of the matrices $C_\alpha$ and the proposed $C_\alpha^{-1}$ is given by:

$$
\sum_{n=0}^{\infty} x^n \left[ \sum_{k=0}^{n} r_k \alpha^k \tilde{r}_{n-k} \alpha^{n-k} \right] = (1 - \alpha x)^{1/2} \times (1 - \alpha x)^{-1/2} = 1, \tag{26}
$$

implying that $\tilde{r}_i \alpha^i$ are indeed the coefficients of $C_\alpha^{-1}$, which concludes the proof. $\square$

**Lemma 7** (Bounds on diagonal entries of $C_{1,\beta}^{-1}$). *The diagonal elements of the inverse square root of the momentum matrix $C_{1,\beta}^{-1}$ defined in equation (9) with parameter $0 \leq \beta < 1$, denoted as $(1, \tilde{c}_1^{1,\beta}, \tilde{c}_2^{1,\beta}, \ldots, \tilde{c}_{n-1}^{1,\beta})$, satisfy the following inequality:*

$$
\tilde{r}_k (1 + \beta) \leq \tilde{c}_k^{1,\beta} \leq 0, \quad \text{for } k \geq 1. \tag{27}
$$

*Proof.* The values $\tilde{c}_k^{1,\beta}$ are given by the convolution of $\tilde{r}_k$ and $\beta^k \tilde{r}_k$:

$$
\tilde{c}_k^{1,\beta} = \sum_{j=0}^{k} \tilde{r}_j \tilde{r}_{k-j} \beta^j = (1 + \beta^k) \tilde{r}_k + \sum_{j=1}^{k-1} \tilde{r}_j \tilde{r}_{k-j} \beta^j. \tag{28}
$$

Since $\tilde{r}_j$ is negative for $j \geq 1$, the summation term is positive. Furthermore, $1 + \beta^k \leq 1 + \beta$, and since $\tilde{r}_k$ is negative, we obtain the lower bound:

$$
\tilde{c}_k^{1,\beta} \geq \tilde{r}_k (1 + \beta). \tag{29}
$$

This bound is tight for $k = 1$ as $\tilde{c}_1^{1,\beta} = -\frac{1+\beta}{2}$.

For the upper bound, we first consider the special cases. When $\beta = 0$, we have $\tilde{c}_k^{1,0} = \tilde{r}_k < 0$. For $\beta = 1$, we formally obtain:

$$\tilde{c}_k^{1,1} = \sum_{j=0}^{k} \tilde{r}_j \tilde{r}_{k-j} = \begin{cases} 1, & k = 0, \\ -1, & k = 1, \\ 0, & \text{otherwise.} \end{cases} \tag{30}$$

This follows from the observation that $C_1^{-1} \times C_1^{-1} = A_1^{-1}$, which has the structure described in the equation.

Since the inequality holds for $k = 1$, we now consider $k \geq 2$, where $\tilde{c}_k^1 = 0$. We show the following, which establishes the upper bound:

**Proposition 1** (Monotonicity of diagonal elements of $C_{1,\beta}^{-1}$)**.** *Let $\tilde{c}_k^{1,\beta}$ be the diagonal elements of $C_{1,\beta}^{-1}$ defined in equation* (9)*. Then $\tilde{c}_k^{1,\beta}$ is an increasing function of $\beta$, varying from $\tilde{r}_k$ at $\beta = 0$ to $0$ at $\beta = 1$.*

*Proof.* To do so, we differentiate $\tilde{c}_k^{1,\beta}$ with respect to $\beta$:

$$\frac{d\tilde{c}_k^{1,\beta}}{d\beta} = k\tilde{r}_k\beta^{k-1} + \sum_{j=1}^{k-1} \tilde{r}_j \tilde{r}_{k-j} j \beta^{j-1} = \beta^{k-1}\left( k\tilde{r}_k + \sum_{j=1}^{k-1} \tilde{r}_j \tilde{r}_{k-j} j \beta^{j-k} \right). \tag{31}$$

To prove that this expression is positive, we analyze the term in brackets. As $\beta \to 0$, the term tends to positive infinity since $\tilde{r}_j \tilde{r}_{k-j}$ are positive and $j - k$ is negative. Moreover, this term is decreasing as $\beta \to 1$, so it suffices to check its non-negativity at $\beta = 1$, i.e.,

$$\left. \frac{d\tilde{c}_k^{1,\beta}}{d\beta} \right|_{\beta=1} \geq 0 \tag{32}$$

Setting $\beta = 1$ in equation (31), we have

$$\left. \frac{d\tilde{c}_k^{1,\beta}}{d\beta} \right|_{\beta=1} = k\tilde{r}_k + \sum_{j=1}^{k-1} \tilde{r}_j \tilde{r}_{k-j} j. \tag{33}$$

To show this, we use an auxiliary identity for the values $\tilde{r}_j j$:

$$\tilde{r}_j j = -\frac{r_j j}{2j-1} = \frac{-1}{2}\left( r_j + \frac{r_j}{2j-1} \right) = \frac{-r_j}{2} + \frac{\tilde{r}_j}{2}. \tag{34}$$

Using the identity (34) in equation (33), we obtain:

$$\begin{aligned} \left. \frac{d(\tilde{c}_k^{1,\beta})}{d\beta} \right|_{\beta=1} &= \frac{1}{2}\tilde{r}_k - \frac{1}{2}r_k + \frac{1}{2}\sum_{j=1}^{k-1} \tilde{r}_j \tilde{r}_{k-j} - \frac{1}{2}\sum_{j=1}^{k-1} r_j \tilde{r}_{k-j} \\ &= \frac{1}{2}\sum_{j=0}^{k} \tilde{r}_j \tilde{r}_{k-j} - \frac{1}{2}\sum_{j=0}^{k} r_j \tilde{r}_{k-j} = 0. \end{aligned} \tag{35}$$

Since both sums vanish for $k \geq 2$, this concludes the proof of Proposition 1. $\qquad \square$

This completes the proof of the lemma. $\qquad \square$

**Lemma 8** (Decresing values)**.** *The values $(1, c_1^{\alpha,\beta}, \ldots, c_{p-1}^{\alpha,\beta}, g_p^{\alpha,\beta}, \ldots, g_{n-1}^{\alpha,\beta})$ of matrix $C_{\alpha,\beta}^p$ as defined in Lemma 12 are decreasing.*

*Proof.* The first $p$ values are decreasing, as shown in Kalinin & Lampert (2024). For the remaining values, we prove that

$$g_{p+k}^{\alpha,\beta} - g_{p+k+1}^{\alpha,\beta} = \sum_{j=1}^{p-1} (-\tilde{c}_j^{\alpha,\beta})(g_{p+k-j}^{\alpha,\beta} - g_{p+k+1-j}^{\alpha,\beta}) \geq 0. \tag{36}$$

In Lemma 7, we prove that $(-\tilde{c}_j^{\alpha,\beta}) \geq 0$, so each term in the summation is non-negative. Since the differences $(g_i^{\alpha,\beta} - g_{i+1}^{\alpha,\beta})$ are also non-negative by the induction step, the inequality follows, completing the proof. $\square$

**Lemma 9** (Bound on the matrix diagonal sum of $C_{1,\beta}^{-1}$)**.** *The diagonal elements of the inverse square root of the momentum matrix $C_{1,\beta}^{-1}$ defined in equation* (9) *with parameter $0 \leq \beta < 1$, denoted as $(1, \tilde{c}_1^{1,\beta}, \tilde{c}_2^{1,\beta}, \ldots, \tilde{c}_{n-1}^{1,\beta})$, satisfy the following inequality:*

$$r_k(1-\beta) \leq \sum_{j=0}^{k} \tilde{c}_j^{1,\beta} \leq c_k^{1,\beta}(1-\beta), \quad \text{for } k \geq 1. \tag{37}$$

*Here $\tilde{c}_i^{1,\beta}$ is as defined by equation* (9).

*Proof.* We first state several properties of the sums of $\tilde{c}_j^{1,\beta}$:

$$(1) \quad \sum_{j=0}^{k} \tilde{c}_j^{1,\beta} = \sum_{j=0}^{k} \tilde{r}_j \beta^j r_{k-j},$$

$$(2) \quad \sum_{j=0}^{k} \tilde{c}_j^{1,\beta} \beta^{k-j} = \sum_{j=0}^{k} r_j \beta^j \tilde{r}_{k-j}, \tag{38}$$

$$(3) \quad \sum_{j=0}^{k} \frac{\tilde{c}_j^{1,\beta}(1 - \beta^{k-j+1})}{1 - \beta} = c_k^{1,\beta}$$

which can be derived from equating the coefficients of the following generating function identities, respectively:

$$(1) \quad \left[\sqrt{1-x}\sqrt{1-\beta x}\right] \times \frac{1}{1-x} = \frac{\sqrt{1-\beta x}}{\sqrt{1-x}}$$

$$(2) \quad \left[\sqrt{1-x}\sqrt{1-\beta x}\right] \times \frac{1}{1-\beta x} = \frac{\sqrt{1-x}}{\sqrt{1-\beta x}} \tag{39}$$

$$(3) \quad \left[\sqrt{1-x}\sqrt{1-\beta x}\right] \times \left[\frac{1}{1-\beta x}\frac{1}{1-x}\right] = \frac{1}{\sqrt{1-x}\sqrt{1-\beta x}}$$

**Upper Bound.** First, we rewrite the expression as follows by multiplying and dividing by $1 - \beta$ the terms $\tilde{c}_j^{1,\beta}$:

$$\sum_{j=0}^{k} \tilde{c}_j^{1,\beta} - c_k^{1,\beta}(1-\beta) = (1-\beta)\sum_{j=0}^{k} \frac{\tilde{c}_j^{1,\beta}(1 - \beta^{k-j+1} + \beta^{k-j+1})}{1-\beta} - c_k^{1,\beta}(1-\beta)$$

$$= \beta \sum_{j=0}^{k} \tilde{c}_j^{1,\beta} \beta^{k-j} = \beta \sum_{j=0}^{k} r_j \beta^j \tilde{r}_{k-j} = \beta^{k+1} \sum_{j=0}^{k} \tilde{r}_j \beta^{-j} r_{k-j}, \tag{40}$$

where the third equality follows from equation 38 (2). For $\beta = 0$, the expression is identically 0. So, now consider when $\beta > 0$. We want to show that

$$\sum_{j=0}^{k} \tilde{r}_j \beta^{-j} r_{k-j} \geq 0 \tag{41}$$

for all $\beta \in (0, 1]$.

As $\beta$ increases from 0 to 1, the sum is clearly increasing, since the only positive term does not have a $\beta$ multiplier. For $\beta = 1$, the sum equals zero, as the sequences $\tilde{r}_j$ and $r_j$ have inverse generating functions. Therefore, the sum remains negative, concluding the proof of the upper bound.

**Lower Bound.** For the lower bound, using equation (38) and the recurrence relation of $\tilde{r}_j$ stated in Lemma 1, we get

$$
\begin{aligned}
\sum_{j=0}^{k} \tilde{c}_j^{1,\beta} - r_k(1-\beta) &= \sum_{j=0}^{k} \tilde{r}_j \beta^j r_{k-j} - r_k(1-\beta) = \sum_{j=1}^{k} \tilde{r}_j \beta^j r_{k-j} + \beta r_k \\
&= \beta \left[ r_k - \sum_{j=1}^{k} \frac{r_j}{2j-1} r_{k-j} \beta^{j-1} \right] \geq \beta \left[ r_k - \sum_{j=1}^{k} \frac{r_j}{2j-1} r_{k-j} \right] \quad (42) \\
&\geq \beta \sum_{j=0}^{k} \tilde{r}_j r_{k-j} = 0,
\end{aligned}
$$

concluding the proof. In the above, the first inequality follows from the fact that $0 < \beta \leq 1$. The fact is trivially true for $\beta = 0$. $\square$

**Lemma 10** (Bound on diagonal values of the matrix $C_{1,\beta}$). *The diagonal values of the matrix $C_{1,\beta}$ (see equation (11)) with parameter $0 \leq \beta < 1$, denoted as $(1, c_1^{1,\beta}, c_2^{1,\beta}, \ldots, c_{n-1}^{1,\beta})$, satisfy the inequality:*

$$
c_k^{1,\beta} \leq c_{k-1}^{1,\beta} \left[ 1 - \frac{(1-\beta)^2}{2k} \right] \quad for \quad k \geq 1. \tag{43}
$$

*Proof.* We first prove that

$$
c_{k-1}^{1,\beta} - c_k^{1,\beta} \geq (r_{k-1} - r_k)(1-\beta). \tag{44}
$$

Using the expression of $c_k^{1,\beta}$, we have

$$
\begin{aligned}
c_{k-1}^{1,\beta} - c_k^{1,\beta} - (r_{k-1} - r_k)(1-\beta) &= \sum_{j=0}^{k-1} r_j r_{k-1-j} \beta^j - \sum_{j=0}^{k} r_j r_{k-j} \beta^j - (r_{k-1} - r_k)(1-\beta) \\
&= \beta(r_{k-1} - r_k) + \sum_{j=1}^{k-1} r_j (r_{k-j-1} - r_{k-j}) \beta^j - r_k \beta^k \\
&= \beta^k \left[ \beta^{1-k}(r_{k-1} - r_k) + \sum_{j=1}^{k-1} r_j (r_{k-j-1} - r_{k-j}) \beta^{j-k} - r_k \right]
\end{aligned}
$$

We note that $r_k$ is a decreasing sequence; therefore, the first two terms inside the brackets are positive, and the powers of $\beta$ in front of them are non-positive. Therefore, as a lower bound, we can substitute $\beta = 1$ inside the sum:

$$
\begin{aligned}
c_{k-1}^{1,\beta} - c_k^{1,\beta} - (r_{k-1} - r_k)(1-\beta) &\geq \beta^k \left[ r_{k-1} - r_k + \sum_{j=1}^{k-1} r_j (r_{k-j-1} - r_{k-j}) - r_k \right] \\
&= \beta^k [r_{k-1} - 2r_k + (1 - r_{k-1}) - (1 - 2r_k)] = 0
\end{aligned} \tag{45}
$$

Using this inequality, we obtain:

$$
\begin{aligned}
\frac{c_k^{1,\beta}}{c_{k-1}^{1,\beta}} &= \frac{c_{k-1}^{1,\beta} - (c_{k-1}^{1,\beta} - c_k^{1,\beta})}{c_{k-1}^{1,\beta}} \leq 1 - \frac{r_{k-1} - r_k}{c_{k-1}^{1,\beta}}(1-\beta) \\
&= 1 - \frac{r_{k-1}}{2k} \cdot \frac{1-\beta}{c_{k-1}^{1,\beta}} \leq 1 - \frac{(1-\beta)^2}{2k},
\end{aligned} \tag{46}
$$

concluding the proof. $\square$

**Lemma 11** (Bounds on diagonals of $B_{\alpha,\beta}^p$). *The matrix $B_{\alpha,\beta}^p$ in the BISR factorization is a lower triangular Toeplitz matrix. The values on its diagonals are*

$$(1, c_1^{\alpha,\beta}, c_2^{\alpha,\beta}, \ldots, c_{p-1}^{\alpha,\beta}, b_p^{\alpha,\beta}, \ldots, b_{n-1}^{\alpha,\beta}) \quad where \quad 0 \le b_i^{\alpha,\beta} \le \alpha^i c_{p-1}^{1,\beta/\alpha} \quad for \quad i \ge p \quad (47)$$

*where $c_i^{1,\beta/\alpha}$ for $1 \le i \le p-1$ is as defined in equation* (11) .

*Proof.* The first $p$ values are identical to the square root factorization $c_i^{\alpha,\beta}$ due to the uniqueness of the inverse. The remaining values satisfy the following recurrence:

$$b_i^{\alpha,\beta} = \sum_{j=0}^{p-1} \tilde{c}_j^{\alpha,\beta} \frac{\alpha^{i-j+1} - \beta^{i-j+1}}{\alpha - \beta} = \alpha^i \sum_{j=0}^{p-1} \tilde{c}_j^{1,\beta/\alpha} \frac{1 - \beta^{i-j+1}}{1 - \beta/\alpha} = \alpha^i b_i^{1,\beta/\alpha}. \quad (48)$$

Therefore, it suffices to prove that $b_i^{1,\beta} \le c_{p-1}^{1,\beta}$, since we can then substitute $\beta$ with $\beta/\alpha$.

$$\begin{aligned} b_i^{1,\beta} &= \sum_{j=0}^{p-1} \tilde{c}_j^{1,\beta} \frac{1 - \beta^{i-j+1}}{1 - \beta} = \frac{1}{1 - \beta} \sum_{j=0}^{p-1} \tilde{c}_j^{1,\beta} - \beta^{i+1-p} \sum_{j=0}^{p-1} \tilde{c}_j^{1,\beta} \frac{\beta^{p-j}}{1 - \beta} \\ &= \frac{1}{1 - \beta} \sum_{j=0}^{p-1} \tilde{c}_j^{1,\beta} + \beta^{i+1-p} \sum_{j=0}^{p-1} \tilde{c}_j^{1,\beta} \frac{(-\beta^{p-j} + 1 - 1)}{1 - \beta} \\ &= \frac{1 - \beta^{i+1-p}}{1 - \beta} \sum_{j=0}^{p-1} \tilde{c}_j^{1,\beta} + c_{p-1}^{1,\beta} \beta^{i+1-p}. \end{aligned} \quad (49)$$

We now use Lemma 9 to first show that $b_i^{1,\beta} \ge 0$, since the sum of $\tilde{c}_j^{1,\beta}$ is non-negative and all other terms are positive. Second, we establish that:

$$b_i^{1,\beta} \le \frac{1 - \beta^{i+1-p}}{1 - \beta}(1 - \beta)c_{p-1}^{1,\beta} + c_{p-1}^{1,\beta} \beta^{i+1-p} = c_{p-1}^{1,\beta}, \quad (50)$$

which completes the proof. $\qquad\square$

**Lemma 12** (Bounds on diagonals of $C_{\alpha,\beta}^p$). *The matrix $C_{\alpha,\beta}^p$ in the BISR factorization is a lower triangular Toeplitz matrix. The values on its diagonals are $(1, c_1^{\alpha,\beta}, c_2^{\alpha,\beta}, \ldots, c_{p-1}^{\alpha,\beta}, g_p^{\alpha,\beta}, \ldots, g_{n-1}^{\alpha,\beta})$, where $c_i^{1,\beta/\alpha}$ (for $1 \le i \le p-1$) is as defined in equation* (11) *and*

$$0 \le g_i^{\alpha,\beta} \le \alpha^i \min\left(c_i^{1,\beta/\alpha}, c_p^{1,\beta/\alpha} \gamma_{\beta/\alpha}^{i-p}\right) \quad for \quad \gamma_{\beta/\alpha} = \left(1 + \frac{(1 - \beta/\alpha)^2}{4p(1 + \beta/\alpha)}\right)^{-1} \quad and \quad i \ge p. \quad (51)$$

*Proof.* The first $p$ values of $C_{\alpha,\beta}^p$ are the same as those of $C_{\alpha,\beta}$ since the matrix is Lower Triangular Toeplitz (LTT). For the subsequent values, we first prove the following inequality by induction:

$$g_i^{\alpha,\beta} = \sum_{j=1}^{p-1} (-\tilde{c}_j^{\alpha,\beta}) g_{i-j}^{\alpha,\beta} \le \sum_{j=1}^{p-1} (-\tilde{c}_j^{\alpha,\beta}) c_{i-j}^{\alpha,\beta} \le \sum_{j=1}^{i} (-\tilde{c}_j^{\alpha,\beta}) c_{i-j}^{\alpha,\beta} = c_i^{\alpha,\beta} = \alpha^i c_i^{1,\beta/\alpha}. \quad (52)$$

We observe that for all sequences $c_i^{\alpha,\beta}$, $\tilde{c}_i^{\alpha,\beta}$, and $g_i^{\alpha,\beta}$, we can factor out $\alpha^i$ by replacing $\beta$ with $\beta/\alpha$. Therefore, it suffices to prove the inequality $g_i^{1,\beta} \le c_p^{1,\beta} \gamma_\beta^{i-p}$, after which we may substitute $\beta$ with $\beta/\alpha$. For the subsequent $p$ values, we establish the stated bound $g_i^{1,\beta} \le c_p^{1,\beta} \gamma_\beta^{i-p}$ using Lemma 10.

$$\frac{g_{p+k}^{1,\beta}}{c_p^{1,\beta} \gamma_\beta^k} \le \frac{c_{p+k}^{1,\beta}}{c_p^{1,\beta}} \left(1 + \frac{(1 - \beta)^2}{4p(1 + \beta)}\right)^k = \prod_{j=1}^{k} \left(1 - \frac{(1 - \beta)^2}{2(p + j)}\right)\left(1 + \frac{(1 - \beta)^2}{4p(1 + \beta)}\right) \le 1. \quad (53)$$

Since each term in the product is less than 1 for $2p + 2j \leq 4p$, the inequality holds. For the induction step, we show:

$$\frac{g_{p+k}^{1,\beta}}{c_p^{1,\beta}\gamma_\beta^k} = \frac{1}{c_p^{1,\beta}\gamma_\beta^k}\sum_{j=1}^{p-1}(-\tilde{c}_j^{1,\beta})g_{p+k-j}^{1,\beta} \leq \sum_{j=1}^{p-1}(-\tilde{c}_j^{1,\beta})\gamma_\beta^{-j} = \sum_{j=1}^{p-1}(-\tilde{c}_j^{1,\beta})\left(1 + \frac{(1-\beta)^2}{4p(1+\beta)}\right)^j. \tag{54}$$

For convenience, we denote $\phi_\beta = \frac{(1-\beta)^2}{1+\beta} < 1$. To proceed, we use the following auxiliary inequality for $j \leq p - 1$:

$$\left(1 + \frac{\phi_\beta}{4p}\right)^j \leq e^{\frac{j\phi_\beta}{4p}} \leq 1 + \frac{5j\phi_\beta}{16p}, \tag{55}$$

since $e^x \leq 1 + 1.25x$ for $x \leq \frac{1}{4}$. Combining this inequality with Lemma 9, we obtain:

$$\frac{g_{p+k}^{1,\beta}}{c_p^{1,\beta}\gamma_\beta^k} \leq \sum_{j=1}^{p-1}(-\tilde{c}_j^{1,\beta})\left(1 + \frac{5j\phi_\beta}{16p}\right) \leq 1 - r_{p-1}(1-\beta) + \frac{5\phi_\beta}{16p}\sum_{j=1}^{p-1}(-\tilde{c}_j^{1,\beta})j. \tag{56}$$

By Lemma 7, we can upper bound:

$$(-\tilde{c}_j^{1,\beta})j \leq (-\tilde{r}_j)j(1+\beta) = \frac{jr_j}{2j-1}(1+\beta) \leq r_j(1+\beta). \tag{57}$$

Using the known bounds $\frac{1}{\sqrt{\pi(j+1)}} \leq r_j \leq \frac{1}{\sqrt{\pi j}}$, we conclude:

$$\begin{aligned}
\frac{g_{p+k}^{1,\beta}}{c_p^{1,\beta}\gamma_\beta^k} &\leq 1 - \frac{1-\beta}{\sqrt{\pi p}} + \frac{5(1-\beta)^2}{16p\sqrt{\pi}}\sum_{j=1}^{p-1}\frac{1}{\sqrt{j}} \leq 1 - \frac{1-\beta}{\sqrt{\pi p}} + \frac{5(1-\beta)^2}{16p\sqrt{\pi}}\cdot 2\sqrt{p} \\
&\leq 1 - \frac{1-\beta}{\sqrt{\pi p}}\left(1 - \frac{5}{8}(1-\beta)\right) < 1,
\end{aligned} \tag{58}$$

where for the second inequality we used the integral estimate $\sum_{j=1}^{k-1}j^{-1/2} \leq \int_0^k x^{-1/2}dx = 2\sqrt{k}$. Thus, we have shown that $\frac{g_{p+k}^{1,\beta}}{c_p^{1,\beta}\gamma_\beta^k} \leq 1$ for all $k$, which completes the proof. $\qquad\square$

**Theorem 4** (BISR Approximation Error). *For $1 \leq p \leq n$ and $1 \leq k \leq \frac{n}{b}$ the following upper bound holds for the matrix factorization error of the BISR $A_{\alpha,\beta} = B_{\alpha,\beta}^p C_{\alpha,\beta}^p$ (as in Definition 1):*

$$\mathcal{E}(B_{\alpha,\beta}^p, C_{\alpha,\beta}^p) = \begin{cases} O_\beta\left(\sqrt{k}\log p + \sqrt{\frac{nk}{b}} + \sqrt{\frac{nk\log p}{p}} + \sqrt{\frac{kp\log p}{b}}\right) & \text{for} \quad \alpha = 1, \\ O_{\alpha,\beta}(\sqrt{k}) & \text{for} \quad \alpha < 1, \end{cases} \tag{7}$$

*where $O_{\alpha,\beta}$ and $O_\beta$ indicate that the hidden constant may depend on $\alpha$ and $\beta$, respectively.*

*Proof.* We begin with the case $\alpha < 1$. To analyze this, we first consider the Frobenius norm:

$$\begin{aligned}
\frac{\|B_{\alpha,\beta}^p\|_{\text{Fr}}^2}{n} &\leq \sum_{i=0}^{p-1}(c_i^{\alpha,\beta})^2 + \sum_{i=p}^{n-1}(b_i^{\alpha,\beta})^2 \leq \sum_{i=0}^{p-1}(c_i^{\alpha,\beta})^2 + (c_{p-1}^{1,\beta/\alpha})^2\sum_{i=p}^{n-1}\alpha^{2i} \\
&\leq \frac{1}{(\alpha-\beta)^2}\log\left(\frac{1}{1-\alpha^2}\right) + \frac{\alpha^{2p}}{1-\alpha^2} = O_{\alpha,\beta}(1),
\end{aligned} \tag{59}$$

where for the second inequality we used Lemma 11, and for the third inequality Lemma 7 from Kalinin & Lampert (2024).

For the $(k,b)$-sensitivity of the matrix $C_{\alpha,\beta}^p$, we use the fact that it is element-wise bounded by the full matrix $C_{\alpha,\beta}$ (see Lemma 12). For $C_{\alpha,\beta}$, we apply a bound from Theorem 7 of Kalinin & Lampert (2024), yielding $\text{sens}_{k,b}(C_{\alpha,\beta}) = O_{\alpha,\beta}(\sqrt{k})$, which concludes the case $\alpha < 1$.

For $\alpha = 1$, we use Lemma 11 to get:

$$\frac{\|B_{1,\beta}^p\|_{\mathrm{Fr}}^2}{n} \leq \sum_{i=0}^{n-1}(b_i^{1,\beta})^2 = \sum_{i=0}^{p-1}(c_i^{1,\beta})^2 + \sum_{i=p}^{n-1}(c_{p-1}^{1,\beta})^2 = \frac{1}{(1-\beta)^2}\sum_{i=0}^{p-1}r_i^2 + \frac{1}{(1-\beta)^2}\sum_{i=p}^{n-1}r_{p-1}^2$$

$$\leq \frac{1}{(1-\beta)^2}\left[1 + \log p + \frac{n-p}{p\pi}\right] = O_\beta\left(\log p + \frac{n}{p}\right).$$

$$(60)$$

Next, we bound the sensitivity under $k, b$ participation. Using Theorem 5, combined with Lemma 8 we obtain:

$$\mathrm{sens}_{k,b}^2(C_{1,\beta}^p) = \sum_{j=0}^{k-1}\sum_{i=0}^{k-1}\langle (C_{1,\beta}^p)_{:,ib}, (C_{1,\beta}^p)_{:,jb}\rangle. \tag{61}$$

We split the sum into the following four terms:

$$\mathrm{sens}_{k,b}^2(C_{1,\beta}^p) = \underbrace{\sum_{i=0}^{k-1}\sum_{j\neq i}^{k-1}\sum_{t=0}^{\min(p+ib,n)-1-jb} c_t^{1,\beta}c_{jb-ib+t}^{1,\beta}}_{\mathcal{S}_1} + \underbrace{\sum_{i=0}^{k-1}\sum_{j\neq i}^{k-1}\sum_{t=0}^{\min(p-1,n-1-jb)} c_t^{1,\beta}g_{jb-ib+t}^{1,\beta}}_{\mathcal{S}_2}$$

$$+ \underbrace{\sum_{i=0}^{k-1}\sum_{j\neq i}^{k-1}\sum_{t=p}^{n-1-jb} g_t^{1,\beta}g_{jb-ib+t}^{1,\beta}}_{\mathcal{S}_3} + \underbrace{\sum_{i=0}^{k-1}\left[\sum_{t=0}^{\min(p-1,n-1-ib)}(c_t^{1,\beta})^2 + \sum_{t=p}^{n-1-ib}(g_t^{1,\beta})^2\right]}_{\mathcal{S}_4}$$

$$(62)$$

**Step 1 ($\mathcal{S}_1$ Bound)** We note that the case $b < p < n$ has not been considered in Kalinin & Lampert (2024) and is technically more challenging. Consider the half of the sum where $j > i$. The sum requires $jb - ib \leq p - 1$; otherwise, the upper limit would be negative. We bound the sum as follows:

$$\sum_{t=0}^{\min(p+ib,n)-1-jb} c_t^{1,\beta}c_{jb-ib+t}^{1,\beta} \leq \frac{r_{jb-ib}}{1-\beta} + \frac{1}{\pi(1-\beta)^2}\sum_{t=1}^{p-1+ib-jb}\frac{1}{\sqrt{t(jb-ib+t)}}$$

$$\leq \frac{1}{(1-\beta)^2}\left[1 + \frac{1}{\pi}\int_0^{p-1+ib-jb}\frac{dx}{\sqrt{x(jb-ib+x)}}\right] \tag{63}$$

$$= \frac{1}{(1-\beta)^2}\left[1 + \frac{1}{\pi}f\left(\frac{jb-ib}{p-1+ib-jb}\right)\right],$$

where $f(a) = 2\log\left(\sqrt{\frac{1}{a}+1} + \sqrt{\frac{1}{a}}\right)$. We then use the following auxiliary inequality for the function $f(a)$:

$$f(a) = \log\left(\frac{1}{a}+1\right) + 2\log\left(1 + \frac{1}{\sqrt{a+1}}\right) \leq \log\left(\frac{1}{a}+1\right) + 2\log 2. \tag{64}$$

This results in the following inequality:

$$\sum_{t=0}^{\min(p+ib,n)-1-jb} c_t^{1,\beta}c_{jb-ib+t}^{1,\beta} \leq \frac{1}{(1-\beta)^2}\left[1 + \frac{2\log 2}{\pi} + \frac{1}{\pi}\log\left(\frac{p-1+ib-jb}{jb-ib}\right)\right]\mathbb{1}_{jb-ib\leq p-1}. \tag{65}$$

We can now upper bound the double sum:

$$\sum_{i=0}^{k-1}\sum_{j\neq i}^{k-1}\sum_{t=0}^{\min(p+ib,n)-1-jb} c_t^{1,\beta}c_{jb-ib+t}^{1,\beta} \leq \frac{2}{(1-\beta)^2}\sum_{i=0}^{k-1}\sum_{j=i+1}^{\min(k-1,i+\lfloor\frac{p-1}{b}\rfloor)}\left[\frac{3}{2} + \frac{1}{\pi}\log\left(\frac{p-1}{jb-ib}\right)\right]. \tag{66}$$

The first term gives us:

$$\frac{2}{(1-\beta)^2} \sum_{i=0}^{k-1} \sum_{j=i+1}^{\min(k-1,i+\lfloor\frac{p-1}{b}\rfloor)} \frac{3}{2} \leq \frac{3k}{(1-\beta)^2} \left\lfloor \frac{p-1}{b} \right\rfloor. \tag{67}$$

The second term is more involved. First, we upper bound the upper limit of the sum $\min(k-1, i + \lfloor\frac{p-1}{b}\rfloor)$ by $i + \lfloor\frac{p-1}{b}\rfloor$, since the summands are positive. We can then upper bound the expression by:

$$\sum_{i=0}^{k-1} \sum_{j=i+1}^{i+\lfloor\frac{p-1}{b}\rfloor} \log\left(\frac{\frac{p-1}{b}}{j-i}\right) = \log \prod_{i=0}^{k-1} \frac{(\frac{p-1}{b})^{\lfloor\frac{p-1}{b}\rfloor}}{(\lfloor\frac{p-1}{b}\rfloor)!} = k \log \frac{(\frac{p-1}{b})^{\lfloor\frac{p-1}{b}\rfloor}}{(\lfloor\frac{p-1}{b}\rfloor)!}. \tag{68}$$

Using the auxiliary inequality $k! \geq (\frac{k}{e})^k$, we show that:

$$\log \frac{(\frac{p-1}{b})^{\lfloor\frac{p-1}{b}\rfloor}}{(\lfloor\frac{p-1}{b}\rfloor)!} \leq \left\lfloor\frac{p-1}{b}\right\rfloor \log\frac{p-1}{b} - \left\lfloor\frac{p-1}{b}\right\rfloor \log\left\lfloor\frac{p-1}{b}\right\rfloor + \left\lfloor\frac{p-1}{b}\right\rfloor$$
$$= \left\lfloor\frac{p-1}{b}\right\rfloor \log\left\{\frac{p-1}{b}\right\} + \left\lfloor\frac{p-1}{b}\right\rfloor \leq \left\lfloor\frac{p-1}{b}\right\rfloor. \tag{69}$$

Resulting in:

$$\sum_{i=0}^{k-1} \sum_{j\neq i}^{k-1} \sum_{t=0}^{\min(p+ib,n)-1-jb} c_t^{1,\beta} c_{jb-ib+t}^{1,\beta} \leq \frac{1}{(1-\beta)^2}\left(3k\left\lfloor\frac{p-1}{b}\right\rfloor + \frac{2}{\pi}k\left\lfloor\frac{p-1}{b}\right\rfloor\right)$$
$$\leq \frac{4k}{(1-\beta)^2}\left\lfloor\frac{p-1}{b}\right\rfloor, \tag{70}$$

which concludes this part of the calculations.

**Step 2 (Bound $\mathcal{S}_2$).** We can bound the inner sum as follows, assuming that $jb - ib \geq p$:

$$\sum_{t=0}^{\min(p-1,n-1-jb)} c_t^{1,\beta} g_{jb-ib+t}^{1,\beta} \leq c_p^{1,\beta} \sum_{t=0}^{p-1} c_t^{1,\beta} \gamma_\beta^{jb-ib+t-p} = c_p^{1,\beta} \gamma_\beta^{jb-ib-p} \sum_{t=0}^{p-1} c_t^{1,\beta} \gamma_\beta^t$$
$$\leq c_p^{1,\beta} \gamma_\beta^{jb-ib-p} \sum_{t=0}^{p-1} c_t^{1,\beta} \leq \frac{r_p \gamma_\beta^{jb-ib-p}}{(1-\beta)^2}\left(1 + \frac{1}{\sqrt{\pi}}\sum_{t=1}^{p-1}\frac{1}{\sqrt{t}}\right) \tag{71}$$
$$\leq \frac{r_p \gamma_\beta^{jb-ib-p}}{(1-\beta)^2}\left(1 + \frac{2\sqrt{p}}{\sqrt{\pi}}\right) \leq \frac{3\gamma_\beta^{jb-ib-p}}{(1-\beta)^2}$$

For our specific choice of $\gamma_\beta = 1 - \frac{\phi_\beta}{4p+\phi_\beta} = \left(1 + \frac{\phi_\beta}{4p}\right)^{-1}$, where $\phi_\beta = \frac{(1-\beta)^2}{1+\beta}$. We rewrite the bound using the following auxiliary inequality:

$$\gamma_\beta^{-p} = \left(1 + \frac{\phi_\beta}{4p}\right)^p \leq e^{\phi_\beta/4} \leq e^{1/4} \leq \frac{4}{3}, \tag{72}$$

This yields the upper bound for the whole sum:

$$\sum_{i=0}^{k-1} \sum_{j\neq i}^{k-1} \sum_{t=0}^{\min(p-1,n-1-jb)} c_t^{1,\beta} g_{jb-ib+t}^{1,\beta} \leq \frac{8}{(1-\beta)^2} \sum_{i=0}^{k-1} \sum_{j=i+1}^{k-1} \gamma_\beta^{jb-ib} \leq \frac{8k\gamma_\beta^b}{(1-\beta)^2(1-\gamma_\beta^b)} \tag{73}$$

We bound $\gamma_\beta^b$ in the following way:

$$\gamma_\beta^b = \left(1 - \frac{\phi_\beta}{4p+\phi_\beta}\right)^b \leq e^{-\frac{b\phi_\beta}{4p+\phi_\beta}} = e^{-\frac{b\phi_\beta}{p}\frac{p}{4p+1}} \leq e^{-\frac{b\phi_\beta}{5p}} \tag{74}$$

Thus,

$$
\begin{aligned}
\sum_{i=0}^{k-1}\sum_{j\neq i}^{k-1}\sum_{t=0}^{\min(p-1,n-1-jb)} c_t^{1,\beta} g_{jb-ib+t}^{1,\beta} &\leq \frac{8k}{(1-\beta)^2(\gamma_\beta^{-b}-1)} \\
&\leq \frac{8k}{(1-\beta)^2(e^{\frac{b\phi_\beta}{5p}}-1)} \leq \frac{40kp(1+\beta)}{b(1-\beta)^4}.
\end{aligned}
\tag{75}
$$

**Step 3 (Bound $\mathcal{S}_3$)** We first bound the inner sum, assuming $j > i$:

$$
\begin{aligned}
\sum_{t=p}^{n-1-jb} g_t^{1,\beta} g_{jb-ib+t}^{1,\beta} &\leq (c_p^{1,\beta})^2 \sum_{t=p}^{n-1-jb} \gamma_\beta^{t-p}\gamma_\beta^{jb-ib+t-p} \\
&\leq \frac{(c_p^{1,\beta})^2\gamma_\beta^{jb-ib}}{1-\gamma_\beta^2} \leq \frac{(c_p^{1,\beta})^2\gamma_\beta^{jb-ib}(4p+\phi_\beta)}{2\phi_\beta} \\
&\leq \frac{r_p^2(4p+1)\gamma_\beta^{jb-ib}}{2\phi_\beta(1-\beta)^2} \leq \frac{5r_p^2\gamma_\beta^{jb-ib}p}{2\phi_\beta(1-\beta)^2} \\
&\leq \frac{5\gamma_\beta^{jb-ib}}{2\pi\phi_\beta(1-\beta)^2} \leq \frac{\gamma_\beta^{jb-ib}(1+\beta)}{(1-\beta)^4}.
\end{aligned}
\tag{76}
$$

This yields the upper bound:

$$
\sum_{i=0}^{k-1}\sum_{j\neq i}^{k-1}\sum_{t=p}^{n-1-jb} g_t^{1,\beta} g_{jb-ib+t}^{1,\beta} \leq \frac{2(1+\beta)}{(1-\beta)^4}\sum_{i=0}^{k-1}\sum_{j=i+1}^{k-1}\gamma_\beta^{jb-ib} \leq \frac{10kp(1+\beta)^2}{b(1-\beta)^6}.
\tag{77}
$$

analogously to the previous step.

**Step 4 (Bound $\mathcal{S}_4$)** We bound the sum of squared column norms as follows:

$$
\begin{aligned}
\sum_{i=0}^{k-1}\left[\sum_{t=0}^{\min(p-1,n-1-ib)}(c_t^{1,\beta})^2 + \sum_{t=p}^{n-1-ib}(g_t^{1,\beta})^2\right] &\leq \frac{k}{(1-\beta)^2}\left[\sum_{t=0}^{p-1}r_t^2 + \frac{r_p^2}{1-\gamma_\beta^2}\right] \\
&\leq \frac{k}{(1-\beta)^2}\left[1+\log p + \frac{5(1+\beta)}{2\pi(1-\beta)^2}\right].
\end{aligned}
\tag{78}
$$

**Step 5 (Combination)** Combining all steps together, we bound the $k, b$ sensitivity as follows:

$$
\begin{aligned}
\mathrm{sens}_{k,b}^2(C_{1,\beta}^p) = \sum_{j=0}^{k-1}\sum_{i=0}^{k-1}\langle (C_{1,\beta}^p)_{:,ib}, (C_{1,\beta}^p)_{:,jb}\rangle \\
&\leq \frac{k}{(1-\beta)^2}\left(1+\log p + \frac{5(1+\beta)}{2\pi} + 10\frac{p(1+\beta)}{b(1-\beta)^2}\left(4 + \frac{1+\beta}{(1-\beta)^2}\right) + 4\left\lfloor\frac{p-1}{b}\right\rfloor\right) \\
&\leq \frac{k(1+\beta)^2}{(1-\beta)^6}\left(2+\log p + 54\frac{p}{b}\right)
\end{aligned}
\tag{79}
$$

Thus,

$$
\mathcal{E}(B_{1,\beta}^p, C_{1,\beta}^p)^2 \leq \frac{k(1+\beta)^2}{(1-\beta)^8}\left(1+\log p + \frac{n-p}{p\pi}\right)\left(2+\log p + 54\frac{p}{b}\right)
\tag{80}
$$

And

$$
\mathcal{E}(B_{1,\beta}^p, C_{1,\beta}^p) = O_\beta\left(\sqrt{k}\log p + \sqrt{\frac{nk}{b}} + \sqrt{\frac{nk\log p}{p}} + \sqrt{\frac{kp\log p}{b}}\right).
\tag{81}
$$

$\square$

**Lemma 3** (Optimal Band Inversion Error). *Let the matrix $C_\lambda^{-1} = \mathrm{LTT}(1, -\lambda, 0, \ldots, 0)$ be a lower triangular Toeplitz matrix with $1$ on the main diagonal and $-\lambda$ on the subdiagonal. Then, for a single participation and the prefix sum matrix $A_{1,0}$, the following bound on the matrix factorization error holds:*

$$\inf_{\lambda \in (0,1)} \mathcal{E}\left(A_{1,0} C_\lambda^{-1}, C_\lambda\right) = O(n^{1/4}). \tag{10}$$

*Proof.* If the matrix $C_\lambda^{-1}$ is given by $\mathrm{LTT}(1, -\lambda, 0, \ldots, 0)$, then its inverse is $C_\lambda = \mathrm{LTT}(1, \lambda, \lambda^2, \ldots, \lambda^{n-1})$. The product $A_{1,0} C_\lambda^{-1} = \mathrm{LTT}(1, 1 - \lambda, \ldots, 1 - \lambda)$, which leads to the following error:

$$\mathcal{E}(A_{1,0} C_\lambda^{-1}, C_\lambda)^2 = \frac{1}{n}\left(1 + (1-\lambda)^2 (n-1)\right) \sum_{k=0}^{n-1} \lambda^{2k} = \frac{(1 + (1-\lambda)^2(n-1))(1 - \lambda^{2n})}{n(1 - \lambda^2)}. \tag{82}$$

Therefore,

$$\inf_{\lambda \in (0,1)} \mathcal{E}(A_{1,0} C_\lambda^{-1}, C_\lambda)^2 \le \frac{\left(2 - \frac{1}{n}\right)\left(1 - \left(1 - \frac{1}{\sqrt{n}}\right)^n\right)}{\sqrt{n}} \le \frac{2}{\sqrt{n}}, \tag{83}$$

when $\lambda = \sqrt{1 - \frac{1}{\sqrt{n}}}$ as $1 - \lambda \le 1 - (1 - \frac{1}{\sqrt{n}}) = \frac{1}{\sqrt{n}}$. The bound follows. $\qquad\square$

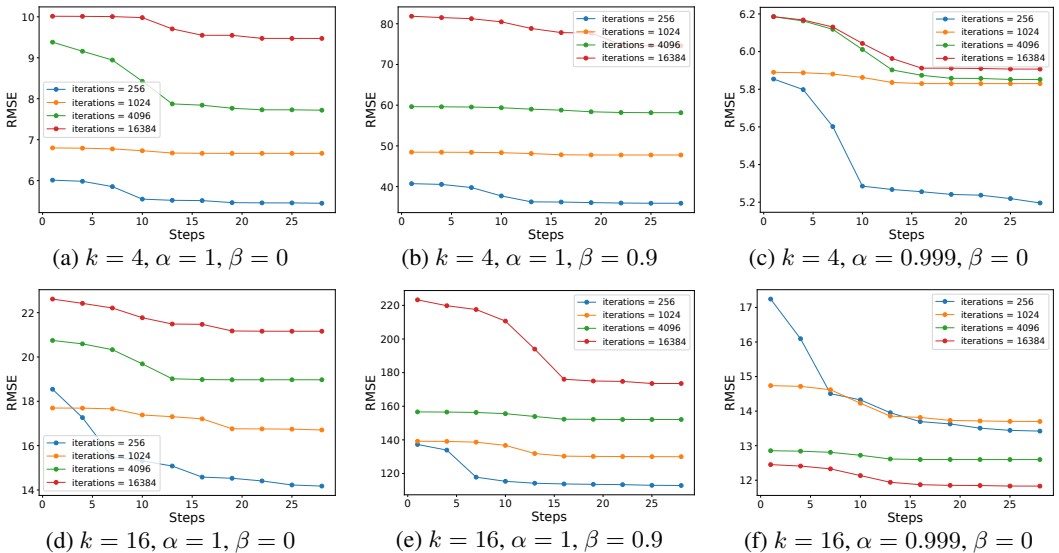

Figure 6: Convergence of Band-Inv-MF under different settings: for participation numbers $k = 4, 16$, with and without momentum ($\beta$) and weight decay ($\alpha$), across various matrix sizes (iterations). In general, we observe that 20 steps are sufficient for the procedure to converge.

Table 1: Hyperparameters for CIFAR-10 Experiments. We train all methods with and without amplification to achieve $(9, 10^{-5})$-differential privacy. Training uses a weight decay of $0.9999$, momentum of $0.9$, and batch size $512$. Noise multipliers are computed via an MCMC accountant for the amplified case, and as $\sigma_{\epsilon,\delta} \times \text{sens}_{k,b}(C)$ for the non-amplified case, assuming 10 training epochs.

|  | Method | Noise Multiplier | Learning Rate | bandwidth | Clip Norm |
|---|---|---|---|---|---|
| Amplified | DP-SGD | 1.2 | 0.1 | 1 | 10 |
|  | BSR | 2.3 | 0.3 | 4 | 10 |
|  | BISR | 4.4 | 0.7 | 4 | 10 |
|  | Band-MF | 2.4 | 0.3 | 4 | 10 |
|  | Band-Inv-MF | 8.2 | 0.4 | 4 | 10 |
| Non-amplified | DP-SGD | 1.8 | 0.1 | 1 | 10 |
|  | BSR | 3.3 | 0.2 | 4 | 10 |
|  | BISR | 5.8 | 0.7 | 4 | 10 |
|  | Band-MF | 3.5 | 0.2 | 4 | 10 |
|  | Band-Inv-MF | 9.1 | 0.5 | 4 | 10 |

## B ADDITIONAL MATERIALS

Table 2: CIFAR-10 experiments with and without amplification, for $\varepsilon = 9$, $\delta = 10^{-5}$ showing test accuracy (%) over 10 epochs. Mean $\pm$ standard error computed over 3 runs.

|  | Method | Epoch 1 | Epoch 2 | Epoch 3 | Epoch 4 | Epoch 5 | Epoch 6 | Epoch 7 | Epoch 8 | Epoch 9 | Epoch 10 |
|---|---|---|---|---|---|---|---|---|---|---|---|
| Amp. | DP-SGD | $12.7 \pm 2.2$ | $28.0 \pm 1.1$ | $34.4 \pm 0.4$ | $37.6 \pm 0.7$ | $39.8 \pm 1.2$ | $41.6 \pm 0.2$ | $42.3 \pm 0.8$ | $42.8 \pm 0.3$ | $43.5 \pm 0.4$ | $44.6 \pm 0.7$ |
|  | BSR | $28.3 \pm 0.7$ | $40.2 \pm 1.1$ | $43.6 \pm 1.1$ | $46.5 \pm 0.9$ | $48.0 \pm 2.0$ | $48.8 \pm 1.4$ | $48.9 \pm 1.4$ | $49.4 \pm 0.7$ | $49.2 \pm 1.2$ | $49.8 \pm 0.3$ |
|  | BISR | $32.3 \pm 0.7$ | $42.7 \pm 1.1$ | $47.5 \pm 1.1$ | $50.3 \pm 0.9$ | $52.8 \pm 2.0$ | $56.5 \pm 1.4$ | $57.9 \pm 1.4$ | $58.5 \pm 0.7$ | $60.5 \pm 1.2$ | $61.8 \pm 0.3$ |
|  | Band-MF | $27.7 \pm 2.0$ | $38.5 \pm 0.3$ | $43.1 \pm 1.6$ | $43.7 \pm 1.8$ | $46.8 \pm 0.8$ | $47.7 \pm 0.3$ | $48.2 \pm 0.6$ | $47.8 \pm 2.6$ | $49.1 \pm 0.6$ | $50.0 \pm 0.4$ |
|  | Band-Inv-MF | $23.6 \pm 2.8$ | $34.6 \pm 1.3$ | $40.0 \pm 2.4$ | $44.6 \pm 1.3$ | $48.6 \pm 1.0$ | $50.4 \pm 1.0$ | $50.6 \pm 0.5$ | $53.4 \pm 0.8$ | $56.2 \pm 0.6$ | $57.4 \pm 1.2$ |
| Non-Amp. | DP-SGD | $19.5 \pm 3.0$ | $31.0 \pm 1.1$ | $36.7 \pm 0.2$ | $37.2 \pm 0.4$ | $37.7 \pm 1.2$ | $39.3 \pm 2.0$ | $39.8 \pm 1.2$ | $39.1 \pm 0.3$ | $39.5 \pm 0.5$ | $39.0 \pm 0.7$ |
|  | BSR | $25.4 \pm 1.2$ | $36.7 \pm 1.2$ | $40.8 \pm 1.1$ | $41.6 \pm 2.0$ | $43.6 \pm 0.9$ | $44.5 \pm 0.7$ | $45.0 \pm 0.9$ | $44.4 \pm 2.1$ | $45.3 \pm 1.8$ | $45.2 \pm 0.8$ |
|  | BISR | $31.8 \pm 1.5$ | $41.7 \pm 2.2$ | $45.4 \pm 1.4$ | $48.5 \pm 1.3$ | $51.1 \pm 1.0$ | $51.4 \pm 2.7$ | $53.8 \pm 1.0$ | $54.0 \pm 1.2$ | $55.5 \pm 0.8$ | $56.2 \pm 0.2$ |
|  | Band-MF | $25.9 \pm 1.5$ | $36.7 \pm 0.9$ | $41.1 \pm 1.4$ | $43.2 \pm 1.3$ | $42.8 \pm 1.4$ | $45.0 \pm 0.2$ | $45.5 \pm 0.4$ | $45.4 \pm 1.8$ | $46.7 \pm 0.9$ | $45.8 \pm 0.2$ |
|  | Band-Inv-MF | $27.4 \pm 3.0$ | $36.0 \pm 2.5$ | $39.5 \pm 2.4$ | $43.7 \pm 1.2$ | $46.7 \pm 0.5$ | $47.0 \pm 2.0$ | $49.7 \pm 1.7$ | $53.5 \pm 0.5$ | $54.4 \pm 1.5$ | $57.9 \pm 0.4$ |

Table 3: Hyperparameters for IMDB Sentiment Analysis Experiments with BERT-base. We train all methods with and without amplification to achieve $(9, 10^{-5})$-differential privacy. Training uses a weight decay of $0.99999$, momentum of $0.95$, and batch size $512$. Noise multipliers are computed via an MCMC accountant for the amplified case, and as $\sigma_{\epsilon,\delta} \times \text{sens}_{k,b}(C)$ for the non-amplified case, assuming 10 training epochs.

|  | Method | Noise Multiplier | Learning Rate | Bandwidth | Clip Norm |
|---|---|---|---|---|---|
|  | DP-SGD | 1.2 | 0.02 | 1 | 10 |
|  | BSR | 2.3 | 0.02 | 4 | 10 |
| Amplified | BISR | 4.4 | 0.15 | 4 | 10 |
|  | Band-MF | 2.4 | 0.02 | 4 | 10 |
|  | Band-Inv-MF | 8.2 | 0.1 | 4 | 10 |
|  | DP-SGD | 1.8 | 0.02 | 1 | 10 |
|  | BSR | 3.3 | 0.02 | 4 | 10 |
| Non-amplified | BISR | 5.8 | 0.15 | 4 | 10 |
|  | Band-MF | 3.5 | 0.02 | 4 | 10 |
|  | Band-Inv-MF | 9.1 | 0.1 | 4 | 10 |

Table 4: IMDB sentiment analysis (BERT-base) with and without amplification, for $\varepsilon = 9$, $\delta = 10^{-5}$ showing test accuracy (%) over 10 epochs. Mean $\pm$ standard error computed over 3 runs.

|  | Method | Epoch 1 | Epoch 2 | Epoch 3 | Epoch 4 | Epoch 5 | Epoch 6 | Epoch 7 | Epoch 8 | Epoch 9 | Epoch 10 |
|---|---|---|---|---|---|---|---|---|---|---|---|
|  | DP-SGD | $71.23 \pm 0.79$ | $82.42 \pm 0.54$ | $84.45 \pm 0.21$ | $85.57 \pm 0.19$ | $86.18 \pm 0.05$ | $86.64 \pm 0.03$ | $86.93 \pm 0.03$ | $86.99 \pm 0.14$ | $87.26 \pm 0.03$ | $87.43 \pm 0.12$ |
|  | BSR | $75.26 \pm 2.67$ | $84.46 \pm 0.25$ | $86.11 \pm 0.29$ | $86.96 \pm 0.11$ | $87.48 \pm 0.13$ | $87.77 \pm 0.07$ | $87.84 \pm 0.08$ | $87.91 \pm 0.13$ | $88.19 \pm 0.14$ | $88.11 \pm 0.06$ |
| Amp. | BISR | $83.27 \pm 0.21$ | $87.08 \pm 0.05$ | $88.16 \pm 0.09$ | $88.83 \pm 0.06$ | $89.20 \pm 0.05$ | $89.21 \pm 0.09$ | $89.49 \pm 0.03$ | $89.41 \pm 0.11$ | $89.60 \pm 0.05$ | $89.58 \pm 0.17$ |
|  | Band-MF | $76.66 \pm 1.24$ | $84.66 \pm 0.17$ | $86.36 \pm 0.17$ | $87.22 \pm 0.15$ | $87.62 \pm 0.40$ | $87.89 \pm 0.23$ | $88.04 \pm 0.19$ | $88.16 \pm 0.21$ | $88.33 \pm 0.26$ | $88.31 \pm 0.19$ |
|  | Band-Inv-MF | $82.91 \pm 0.53$ | $86.92 \pm 0.31$ | $88.25 \pm 0.05$ | $88.60 \pm 0.17$ | $89.02 \pm 0.19$ | $89.23 \pm 0.06$ | $89.50 \pm 0.06$ | $89.50 \pm 0.06$ | $89.65 \pm 0.00$ | $89.53 \pm 0.25$ |
|  | DP-SGD | $62.94 \pm 1.47$ | $78.21 \pm 0.56$ | $83.05 \pm 0.19$ | $84.17 \pm 0.20$ | $85.01 \pm 0.06$ | $85.40 \pm 0.20$ | $85.53 \pm 0.17$ | $85.80 \pm 0.13$ | $85.81 \pm 0.17$ | $85.71 \pm 0.09$ |
|  | BSR | $76.74 \pm 1.04$ | $84.09 \pm 0.09$ | $85.70 \pm 0.22$ | $86.59 \pm 0.26$ | $86.93 \pm 0.03$ | $87.15 \pm 0.13$ | $87.20 \pm 0.11$ | $87.19 \pm 0.09$ | $87.31 \pm 0.15$ | $87.18 \pm 0.10$ |
| Non-Amp. | BISR | $81.85 \pm 0.86$ | $86.65 \pm 0.33$ | $87.60 \pm 0.39$ | $88.64 \pm 0.10$ | $88.90 \pm 0.05$ | $88.83 \pm 0.18$ | $89.02 \pm 0.09$ | $89.43 \pm 0.07$ | $89.37 \pm 0.00$ | $89.42 \pm 0.15$ |
|  | Band-MF | $73.39 \pm 1.39$ | $83.63 \pm 0.38$ | $85.39 \pm 0.16$ | $86.09 \pm 0.29$ | $86.99 \pm 0.06$ | $87.08 \pm 0.13$ | $87.23 \pm 0.16$ | $87.34 \pm 0.14$ | $87.34 \pm 0.07$ | $87.08 \pm 0.14$ |
|  | Band-Inv-MF | $81.99 \pm 0.23$ | $86.06 \pm 0.19$ | $87.98 \pm 0.11$ | $88.46 \pm 0.12$ | $88.63 \pm 0.03$ | $88.79 \pm 0.32$ | $88.85 \pm 0.29$ | $89.02 \pm 0.21$ | $88.89 \pm 0.37$ | $89.23 \pm 0.16$ |

```python
import jax_privacy
from jax_privacy.maxtrix_factorization import toeplitz
import jax.numpy as jnp
import functools
import numpy as np

def expected_mean_error(inv_coef, n, k, workload_coef) -> float:
    inv_coef = jnp.pad(inv_coef, (0, n - inv_coef.size))
    B_norm_squared = toeplitz.mean_error(noising_coef=inv_coef, n=n,
    workload_coef=workload_coef, skip_checks=True)

    coef = toeplitz.inverse_coef(inv_coef)
    min_sep = n // k # assume divisible

    sensitivity_squared = toeplitz.minsep_sensitivity_squared(coef,
    min_sep, k, n, skip_checks=True)

    return sensitivity_squared * B_norm_squared

def compute_square_root(x, n) -> np.ndarray:
    y = np.zeros(n)
    y[0] = np.sqrt(x[0])
    for k in range(1, n):
        y[k] = (x[k] -np.dot(y[1:k], y[1:k][::-1])) / (2 * y[0])
    return y

def init(n, p, alpha = 1.0, beta = 0.0) -> jnp.ndarray:
    x = jnp.array([1, -alpha - beta, alpha * beta] + [0]*(n-3))
    return jnp.array(compute_square_root(x, n)[:p])

def Band_Inv_MF(n, b, k, p, alpha, beta, steps = 20):
    # compute workload matrix
    M = jnp.array([(alpha ** (k + 1) - beta ** (k + 1)) / (beta - alpha)
    for k in range(n)])

    # initialize with BISR coefficients
    C_inv_init = init(n, p, alpha, beta)

    # optimize!
    C_inv_opt = toeplitz.optimize_banded_toeplitz(
      n=n,
      bands=p,
      strategy_coef=C_inv_init,
      loss_fn=functools.partial(expected_mean_error, k=k, workload_coef=M
    ),
      max_optimizer_steps=steps,
    )
    return C_inv_opt
```

Listing 1: Python code for Band-Inv-MF factorization. The function "Band_Inv_MF" takes the matrix size ($n$), the minimum separation ($b$), the number of participations ($k$), the bandwidth ($p$), the weight decay ($\alpha$), the momentum ($\beta$), and the number of optimization steps.

