# OpenReview forum: "Back to Square Roots: An Optimal Bound on the Matrix Factorization Error for Multi-Epoch Differentially Private SGD"
_ICLR.cc/2026/Conference — ICLR 2026 Poster_

### Official Review · Reviewer_xJi2 · 2025-10-26

**Soundness:** 3
**Presentation:** 1
**Contribution:** 4
**Rating:** 8
**Confidence:** 3

**Summary:**

This paper addresses the problem of adding correlated noise to elements in a data stream with a momentum structure in a differentially private manner under continuous observation, while maintaining low approximation error. The authors propose a novel factorization mechanism for injecting correlated noise at each iteration, providing formal differential privacy guarantees even against an adversary with access to all entries of the stream.

Their approach improves upon existing methods by factorizing the workload matrix, extending the square-root factorization framework studied in prior work. This refined factorization yields tighter approximation error bounds. Furthermore, the authors establish matching lower bounds on the achievable approximation error for general classes of stream structures, demonstrating the optimality of their approach.

The proposed mechanism has clear practical relevance, particularly for differentially private stochastic gradient descent (DP-SGD) with momentum. It enables more efficient and accurate private training of deep learning models, highlighting the work’s potential impact on privacy-preserving machine learning.

**Strengths:**

This paper is very strong in terms of results:
1. The paper presents rigorous theoretical results based on novel techniques that effectively close a significant gap in the existing literature on matrix factorization mechanisms within the context of streaming differential privacy covering a large variety of tasks.

2. The paper synthesizes concepts from several related domains to develop a generalized formulation of DP-SGD with momentum, extending beyond existing approaches in the literature.

3. The paper proposes a computationally efficient method to compute correlated noises. It provides a practical technique to implement these correlated noise techniques for settings which are not simple SGD and attempt to break into the practical scenario in existing optimization techniques used for differentially private deep learning.

4. Practically the algorithm performs very well compared to existing matrix factorization methods for DP-SGD on classification datasets.

**Weaknesses:**

The main weakness of this paper lies in its presentation. While the technical results appear sound and potentially impactful, the exposition makes it difficult for readers to fully grasp the scope and implications of the work (especially from the lower bound perspective). Below are some specific comments and suggestions:
1. *Dependence on Prior Work for Context*: The paper relies heavily on Kalinin & Lampert (2024) to set up the background for its lower bound results. For instance, the “multi-participation setting” introduced in Theorem 3 is not defined anywhere in the paper and an appropriate reference has not been given as well, which makes it hard to understand the exact conditions under which the stated bounds apply.

*Suggestion*: It would be helpful if the authors could briefly explain or restate the definition of the multi-participation setting, even if it was introduced in prior work. A short contextual description would make the paper more self-contained and easier to follow.

2. *Ambiguity in Notation*: Some notations are not clearly introduced. In particular, the terms $\Omega_\alpha$ and $\Omega_{\alpha, \beta}$ in Theorem 3, along with their Big-O counterparts in Theorem 4 and Corollary 1, are not defined.

*Suggestion*: Please clarify the meaning of these notations, either directly in the text or in a concise notation table.

Including short background explanations—perhaps in an appendix if not in the main text—would greatly improve the paper’s readability. While it is understandable that some results depend on established frameworks, providing minimal context would help readers appreciate the contributions without needing to refer extensively to other works.

**Questions:**

Please see weaknesses.

---

> ### Author Response · Authors · 2025-11-19
>
> We thank the reviewer for the high evaluation of our paper and for the valuable suggestions.
>
> **>> Dependence on Prior Work for Context**
>
> The multi-participation setting is discussed in the background section. We agree that introducing an explicit definition would improve clarity, and we appreciate the suggestion. We are happy to include it in the revised version.
>
> **>> Ambiguity in Notation**
>
> The notation $O_{\alpha, \beta}$ and $\Omega_{\alpha, \beta}$ extends the usual $O$ and $\Omega$ notation by indicating that the constants may depend on $\alpha$ and $\beta$. In other words, we do not explicitly specify the dependence on the parameters $\alpha$ and $\beta$, although the corresponding lower and upper bounds can be found in the appendix. We are happy to add clarifying remarks in the revised version of the paper.

---

> > ### Comment · Reviewer_xJi2 · 2025-11-26
> >
> > Thank you so much for addressing my concerns! I have gone through the results after the clarification and the results make much more sense. I would like to keep my current score but am more confident about my current score.

---

### Official Review · Reviewer_2RDV · 2025-10-29

**Soundness:** 3
**Presentation:** 3
**Contribution:** 3
**Rating:** 6
**Confidence:** 3

**Summary:**

This paper presents a new algorithm banded inverse square root (BISR), which is an extension to BSR from (Kalinin & Lampert 2024) by imposing a banded structure on the inverse matrix $C^{-1}$ rather than $C$ itself.

By making such modification, the authors are allowed to give a more explicit expression error with respect to the bandwidth $p$ and reduce the overall computational complexity. The authors also show a tight error bound on the approximation error with matching lower bounds.

**Strengths:**

I find the idea of bounding the bandwidth of matrix on matrix $C$ to reduce computational complexity interesting and the optimal error bound a solid contribution. Also, I appreciate the authors' efforts make to make a comprehensive discussion and comparison with prior work.

**Weaknesses:**

1. The algorithmic modification are relatively minimal. The general structure of the algorithm directly follow that in (Kalinin & Lampert 2024) which slightly weaken the contribution of this work.
2. More discussion is needed on the approximation error defined in equation (1) . In particular, how is this error related to the convergence error? Would achieving an optimal bound on this error also imply an optimal convergence rate? It would be great if the authors can provide explicit convergence rates for certain loss function classes (e.g., convex-Lipschitz loss) and make comparisons with existing DP algorithms like DP-SGD.
3. The algorithm, claimed as computational efficient, may not be suitable for modern model training. To achieve optimal rate, $p$ needs to be set as $\tilde{O}(b)$ where $b$, as far as I understand, can be approximated as the number of update steps per epoch. In large-scale training, this number can reach tens of thousands or even millions. Therefore, combining $p$ different $Z_i$'s in each update step could become prohibitively expensive in practice.

**Questions:**

Questions are included in the "Weaknesses" section.

---

> ### Author Response · Authors · 2025-11-19
>
> We thank the reviewer for their review and interesting questions!
>
> **>>  The algorithmic modification are relatively minimal.**
>
> We would like to emphasize that our work exist within a well-established framework of matrix factorization, where  the central question is what correlation matrix to use. However, our contribution is not limited to introducing a single new correlation matrix. We propose an entire class of matrices, called banded inverse matrices, which possess several beneficial properties. These include achieving good RMSE with a relatively small bandwidth, providing strong memory efficiency, and being theoretically easier to analyze in the multi-participation setting.
>
> **>> More discussion is needed on the approximation error defined in equation (1) . In particular, how is this error related to the convergence error?**
>
> For the discussion of the relationship between RMSE and accuracy, we kindly refer to our response to Reviewer 7NNd.
>
> **>> To achieve optimal rate, $p$ needs to be set as $O(b)$ where $b$, as far as I understand, can be approximated as the number of update steps per epoch.**
>
> Thank you for this question. To achieve the lowest possible RMSE for the BISR factorization, we indeed need to set $p = O(b)$, which can be large. We address this issue from two perspectives. If we only care about obtaining a good RMSE and it accurately predicts the overall quality (for instance, in the non-adaptive continual counting), then we introduce BandInvMF. This is an optimized factorization that numerically converges to an almost optimal value within its class for a very small bandwidth (between $4$ and $8$). On the other hand, for deep learning task, we ran the experiments in Figure 4 with bandwidth $p = 4$, showing that BISR can outperform both BandInvMF and banded methods under the same memory constraint, significantly improving over DP-SGD.
>
> It is true, however, that we do not provide theoretical guarantees for memory efficiency. We view this as important future work, although it is also very challenging.

---

### Official Review · Reviewer_53th · 2025-11-01

**Soundness:** 3
**Presentation:** 2
**Contribution:** 2
**Rating:** 6
**Confidence:** 3

**Summary:**

The paper investigates the matrix factorization mechanism for differentially private stochastic gradient descent (DP-SGD) under multi-epoch/multi-participation settings.
The matrix factorization mechanism is used to inject correlated noise into gradients during training by means of a correlation matrix.
The authors propose to enforce a banded structure on the _inverse_ of the correlation matrix, to derive explicit upper bounds on factorization errors and to prove asymptotic optimality.
The authors compare their approach with existing techniques, showing that their proposed approach achieves higher or comparable accuracy for large matrices.
Moreover, the paper proposes an efficient, low-memory method which matches the performance of state-of-the-art approaches while being more efficient.

**Strengths:**

* **Theoretical contribution.**
The paper introduces and discusses a matrix factorization technique with provable optimality, and refines prior existing bound (Kalinin and Lampert, 2024).
Unlike related work, you provide an explicit dependence on the bandwidth $p$ and on the participation $b$, which leads to more useful guarantees.
The idea to consider the inverse correlation matrix instead of the matrix itself is, as far as I can tell, novel and elegant.
Moreover, the discussion on an efficient implementation reinforces the practical utility of your approach.
The problem discussed is very relevant and well placed within related literature.

* **Empirical validation.**
The empirical validation you present is limited but consistent, and qualitatively supports your claims.
In particular in low-resource regimes, the presented approach performs better than existing ones.

**Weaknesses:**

* **Clarity and accessibility.** The paper has dense notation and long proofs: intuition could be introduced earlier.
For instance, the benefits of inverse banding are not intuitively clear, and visualizations could help here.

* **Low privacy regime.**
In your empirical evaluation, you only present results in a arguably low privacy regime $\epsilon=9$.
While this specific value for the privacy budget seems to be common in related literature, it is generally understood to be at the edge of what is considered to be differentially private at all.
If my understanding is correct, the benefits of one specific factorization technique against another, is particularly important when the amount of noise added is small.
While this justifies the chosen privacy regime, it may diminish the contribution for more strict, and therefore relevant from a privacy perspective, privacy regimes ($\epsilon <= 1$) where DP constraints are more meaningful.

* **Empirical relevance.**
Following from my previous point, I am not convinced of the practical relevance of the approach.
While I understand that your contribution is, first of all, theoretical, a comparison with more recent DP-SGD variants would further justify its practical relevance.
The plots do not show any measure of dispersion (e.g., error bars).
The empirical setup does not seem to reflect a real-world application with realistic privacy requirements, and no investigation of the effectiveness of the approach with different privacy budgets is performed.
From the plots, the correspondence between RMSE and accuracy is difficult to grasp.

### Other remarks

* The nomenclature is at times confusing.
Both $C$ and $C^{-1}$ are referred to as "correlation matrix" throughout the introduction (e.g., compare lines 45 and 59/61).

**Questions:**

* How practically (or theoretically) relevant is your approach for stricter privacy regimes, i.e., small values of $\epsilon$?
* What is the standard deviation/variability of the results reported in Figure 1/2? Are the results significantly different if you include error bands in the plots/results?

---

> ### Author Response · Authors · 2025-11-19
>
> We thank the reviewer for their time and interesting questions!
>
> **>> Clarity and accessibility**
>
> Thank you for raising this concern. One of the most important properties of banded inverse matrices is their low memory usage, as even a small bandwidth is sufficient to capture the effect of noise correlation. We can provide a visualization of noise correlation via a banded inverse matrix in the revised version of the paper.
>
> **>> Low privacy regime**
>
> We are happy to provide additional experiments during the discussion period and include them with all necessary details in the revised version. In general, we observed that BISR outperforms other factorizations, at least under a given memory constraint, across a wide range of privacy levels and bandwidths.
>
> **>> Empirical relevance**
>
> We are very sorry for the possible confusion. The plots in Figure 4 are **drawn with error bars**: we used the *fill_between* Matplotlib function to represent the dispersion of the values. This provides the same information as traditional error bars while being, arguably, more visually pleasing. If the question was regarding the Figures 1-3, those are deterministic values and naturally do not have dispersion. Regarding the correspondence between accuracy and RMSE, we will include the values in a table accompanying the corresponding plot. For a more general discussion of the relationship between RMSE and accuracy, we kindly refer to our response to Reviewer 7NNd.
>
> **>> The nomenclature is at times confusing. Both $C$ and $C^{−1}$ are referred to as "correlation matrix" throughout the introduction (e.g., compare lines 45 and 59/61).**
>
> Thank you for pointing this out; the terminology is in fact somewhat ambiguous in the related literature. We will, however, stick to referring to $C^{-1}$ as the correlation matrix, and to
> $C$  as the strategy matrix.
>
> **>> How practically (or theoretically) relevant is your approach for stricter privacy regimes, i.e., small values of $\varepsilon$?**
>
> Theoretically, the RMSE-optimized factorization is always better than DP-SGD. In practice, the situation is more nuanced. However, in additional experiments done after the paper submission, we observed that both BISR and BandInv outperform DP-SGD for $\varepsilon=1$ as well. We would be happy to provide the detailed experimental results during the discussion period.
>
> **>> What is the standard deviation/variability of the results reported in Figure 1/2? Are the results significantly different if you include error bands in the plots/results?**
>
> We kindly refer to our “Emperical relevance” response.

---

> > ### Comment · Reviewer_53th · 2025-11-26
> >
> > Thank you for your reply and for clarifying my concerns. I look forward to see the additional results for stricter privacy regimes you obtained, which I think would strengthen your contribution significantly. I will, until then, keep my score.

---

### Official Review · Reviewer_7NNd · 2025-11-05

**Soundness:** 4
**Presentation:** 4
**Contribution:** 4
**Rating:** 10
**Confidence:** 4

**Summary:**

This paper introduces and analyzes a novel matrix factorization scheme, called BISR, which can be applie to DP-SGD and relies on (i) computing the inverse square root of the workload's matrix, (ii) imposing a band structure on it, and (iii) inverting the resulting matrix again.
A novel lower bound on factorization error for SGD's workload matrix is derived, with matching upper bound for the BISR method, which relies on band structure of the inverse square root.
Numerical experiments show that the resulting factorizations are consistently on par than the existing BSR mechanism, and strongly outperform it for some choices of bandwidth.

**Strengths:**

1. The proposed method, relying on imposing structure on the inverse of SGD's workload matrix squared root, is original and very different from existing ideas.
2. A refined theoretical lower bound on the factorization of SGD's workload matrix is provided.
3. The BISR method is shown to match this upper bound.
4. BISR is shown numerically to provide better factorization in many setting that existing methods, and this improved factorization precision results in improved accuracy over existing method in multiple private machine learning tasks.
5. The paper is very well written and easy to read.

**Weaknesses:**

The paper is very interesting, and I mostly remark strengths about the contributions. Nonetheless, there are some minor weaknessses:
1. No theoretical guarantees for DP-SGD under the BISR matrix factorization are provided.
2. While the theoretical claims hint for a large improvement over the BSR method, this does not always show in practice; studying more precisely (i.e., non-asymptotically) the respective behaviour of the two methods may reveal more subtle compromises.
3. The experiments on CIFAR-10 and IMDB are performed in the small bandwidth and low privacy regime: it is not clear whether the same conclusions would hold in high-privacy/large bandwidth regimes.

**Questions:**

1. Is it possible to derive theoretical convergence guarantees for DP-SGD in simple settings (e.g., strongly-convex functions)? In this setting, is there a chance to observe the true metrics on matrix factorization approximation that impact the final privacy-utility trade-off?
2. Authors claim that the RMSE may not be a good proxy for approximation error. Are there other candidates for better proxies?
3. Experiments showcase the low privacy regime: how would the result change in a high privacy regime? Would BISR still be better? Would the difference increase/decrease?
4. What is the intuition why the inverse square root should be closer to a band structure than the square root itself?

---

> ### Author Response · Authors · 2025-11-19
>
> Thank you for your positive and helpful review. Here we address some of the questions:
>
> **>> 1. Is it possible to derive theoretical convergence guarantees for DP-SGD in simple settings (e.g., strongly-convex functions)? In this setting, is there a chance to observe the true metrics on matrix factorization approximation that impact the final privacy-utility trade-off?**
>
> It is possible, and in fact Matrix Factorization has been analyzed in a variety of settings, including the convex setting (Proposition 4.1 [1]), the non-convex setting (Theorem 4.6 [2]), and linear regression (Theorem 2.2 [3]). In the work by Denisov et al. [1], it was proposed to optimize for RMSE. In the subsequent works [2, 3], optimization-aware metrics were introduced; however, they did not observe numerical improvements for deep learning problems. A very recent work [5] proposed a second-order loss-aware factorization, where the Hessian matrix is computed on publicly available data. In that work, an alternative metric was proposed, which, however, relies on public data with a similar distribution. These proposed metrics were not widely adopted, and the problem of identifying the optimal objective remains open. See the recent survey on related work [4], Section 5.2.
>
> **>> 2. Authors claim that the RMSE may not be a good proxy for approximation error. Are there other candidates for better proxies?**
>
> Following up on our response to the previous question, it may be beneficial to revisit the results of [2] and [3] for our newly proposed class of factorizations, the banded inverse factorizations. In relation to our work, we believe that BISR performs better than BandInvMF because it adds less noise per iteration (lower sensitivity) and is, informally speaking, “closer to the identity matrix than BandInvMF,” while still achieving good noise cancellation, e.g., low RMSE. We consider this an important open question, which can now be approached from a different perspective once we have introduced a new class of matrix factorizations.
>
> **>> 3. Experiments showcase the low privacy regime: how would the result change in a high privacy regime? Would BISR still be better? Would the difference increase/decrease?**
>
> We are happy to provide additional experiments during the discussion period and include them with all necessary details in the revised version. In general, we observed that BISR outperforms other factorizations, at least under a given memory constraint and across a wide range of privacy levels and bandwidths.
>
> **>> 4. What is the intuition why the inverse square root should be closer to a band structure than the square root itself?**
>
> The square root $C$ of the prefix-sum matrix $A$ is a lower-triangular Toeplitz matrix with positive, decreasing coefficients. The inverse matrix $C^{-1}$ can be computed as $A^{-1}C$, where $A^{-1}$ has $1$ on the main diagonal and $-1$ on the subdiagonal and $0$ elsewhere. Taken together, these facts imply that $C^{-1}$ has more rapidly decreasing coefficients on average, since $A^{-1}C$ essentially computes the discrete derivative of the decreasing sequence, producing values that are (on average) smaller than the sequence itself.
>
>
>
>
> [1] Denisov et al. “Improved Differential Privacy for SGD via Optimal Private Linear Operators on Adaptive Streams” NeurIPS 2022
>
> [2] Koloskova et al. “Gradient Descent with Linearly Correlated Noise: Theory and Applications to Differential Privacy” NeurIPS 2023
>
> [3] Choquette-Choo et al. “Correlated Noise Provably Beats Independent Noise for Differentially Private Learning” ICLR 2024
>
> [4] Pillutla et al. “Correlated Noise Mechanisms for Differentially Private Learning” arXiv:2506.08201 2025
>
> [5] Gut et al. “Correlating Cross-Iteration Noise for DP-SGD using Model Curvature” arXiv:2510.05416 2025

---

### Author Response · Authors · 2025-11-30
**High Privacy Experiments**

Following up on the discussion, we are glad to provide a new plot for the high-privacy regime. Please find the plot via the anonymized link:

https://anonymous.4open.science/r/ICLR2026_Rebuttal-0541/amplified_cifar10_accuracy_epsilon_1.pdf


The plot shows CIFAR-10 experiments with a CNN model for $\epsilon = 1$. All factorizations are amplified via Balls-and-Bins amplification. The higher the privacy, the less benefit we observe from optimizing for RMSE. This explains the drop for BandMF and BandInvMF; the latter drops even faster since it has a greater capacity to overfit for RMSE. In contrast, optimization-free methods such as BSR and BISR exhibit the best performance, with our proposed BISR outperforming all alternative methods.

---

### Meta-Review · Area_Chair_K112 · 2025-12-12

**Summary:**

Overall, the reviewers find this paper interesting and useful.

Two reviewers requested experimental results at a lower epsilon (stronger privacy). The authors provided the CIFAR-10 experimental result, where the proposed method outperforms others toward the end of training (at longer epochs). I do not know if the reviewers would have increased the score as a result of seeing the new results. If they focused on whether the proposed method still outperforms others under the stronger privacy regime, maybe they did. However, I am a bit suspicious that the DP-SGD results are almost identical, with accuracy around 40%, when epsilon=1 and epsilon=9 (Figure 4 in the submitted paper). So, if I were one of the reviewers, I probably would have pursued this further and asked whether I could trust their new results.

Reviewer 2RDV mentioned that the general structure of the algorithm directly follows that in (Kalinin & Lampert 2024), which slightly weakens the contribution of this work. I agree with the reviewer in this regard. Unfortunately, there is no way to know whether this reviewer would have changed the score after reading the authors' rebuttal.

Nevertheless, the four reviewers agree that this paper is in good shape and that their contributions are sufficient for publication. Hence, I recommend accepting the paper.

**Reviewer Concerns:**

Two reviewers requested theoretical guarantees for DP-SGD under the BISR matrix factorization. The authors did not show this directly but instead referred to relevant work to demonstrate its existence.

**Reviewer Scores:**

Two reviewers who requested experimental results at a lower epsilon would have increased the score. But even without increased scores, the paper got sufficiently high scores to be accepted.

---

### Decision · Program_Chairs · 2026-01-26

Accept (Poster)